# STABILITY BOUNDS FOR DOMAIN GENERALIZATION UNDER LIMITED DATA

## ABSTRACT

The less-sample learning problem is challenging for machine learning, as it leads to unstable model estimation, i.e., the risk gap between the empirical risk and the expected risk for models increases as the size of the training data decreases. To address this, the classical VC-bound suggests reducing the VC-dimension of models through regularization. However, the data in domain generalization are not independent and identically distributed (i.i.d.), which implies that such bounds fail to provide effective guidance for learning. To fill this gap, we present stability bounds. Specifically, we derive a general exponential-decay upper bound based on the notion of stability for models and McDiarmid's inequality. Based on this, we then present stability bounds for models obtained by regularization-based learning methods. Finally, we apply this result to a classification case and develop a learning method. We also study the stability and generalization error bounds of the proposed learning method, as well as its convergence properties. Additionally, we conduct experiments using datasets with different data sizes to analyze the effectiveness of our methods in real-world applications.

## 1 INTRODUCTION

Domain generalization (DG) aims to train machine learning (ML) models using several available data domains, ensuring they perform well on related but inaccessible (unseen or unknown) domains. This is meaningful because it uses finite sets to induce methods applicable to infinite sets, which can be regarded as a foundation for developing artificial general intelligence. Recently, several studies have been proposed to achieve this goal, such as (Arjovsky et al., 2019; Choraria et al., 2023; Shi et al., 2022). These methods not only offer new perspectives for addressing the domain shift problem but also promote the development of research in DG. For example, based on the minimax risk problem, i.e., empirical risk minimization (ERM), researchers have introduced invariance-based learning objectives (Arjovsky et al., 2019; Ahuja et al., 2020) and causality-based learning objectives (Christiansen et al., 2021; Wang et al., 2023), among others. However, for real-world data scenarios, these methods have their limitations. This is because, under such scenarios, the quantity of data samples is relatively low, sometimes even worse, which is incompatible with the conditions required for discovering invariant or causal features. Namely, the less-sample problem challenges learning methods.

According to the classical VC-bound (Vapnik, 2013), $\varepsilon \leq C\sqrt{d_{vc}/N}$, the risk gap $\varepsilon$ between the empirical risk and the expected risk increases as the size $N$ of the training data decreases, while the VC-dimension $d_{vc}$ is maintained unchanged. This leads to unstable model estimation under limited data conditions. To address the risks associated with limited data samples, a common approach is to incorporate data augmentation techniques to expand $N$. However, there is a significant risk associated with that method: the increased labeling noise rate, as there are no theoretical guarantees for maintaining the same level of labeling noise after applying data augmentation methods. Then, if $N$ cannot be changed, decreasing $d_{vc}$ is the best choice, and Vapnik (2013) suggested using structural risk minimization (SRM) principle, i.e., *regularization*. However, the data are not i.i.d. in domain generalization, which indicates a limitation of using the VC-bound to guide learning.

In this work, we present stability bounds based on McDiarmid's inequality to address the above limitations. But why stability, and why McDiarmid's inequality? First, Bousquet & Elisseeff (2002) provided analyses of the stability of regularization-based learning objectives, which exhibit how

stable the predictions made by the outputs of learning algorithms are across different datasets. Second, the assumption on random variables in McDiarmid's inequality (McDiarmid et al., 1989) only requires independence, and this inequality is more flexible and is capable of estimating nonlinear statistics (Li & Liu, 2024). Our result, $\varepsilon \leq C/\sqrt{\lambda N}$ and $\varepsilon \leq C/\sqrt{\lambda E N}$, shows that regularization-based learning methods can control the penalty factor $\lambda$ to handle different data sizes, thereby obtaining a stable model estimation and reducing the generalization gap $\varepsilon$. If the data amount is small, a large value of $\lambda$ is required to obtain a satisfactory training result, and vice versa. This result aligns with the analyses in (Bousquet & Elisseeff, 2002) and with the suggestion of reducing the $d_{vc}$ by Vapnik (2013). Additionally, we develop a learning method to apply our theoretical results. We also study the stability and generalization error bounds of the proposed learning method, as well as its convergence properties. Based on experimental results from two publicly available real-world datasets, we provide empirical evidence supporting the theoretical results and the proposed learning method. In conclusion, we contribute by (1) Providing the exponential-decay upper bounds for models obtained by regularization-based learning methods in DG. By the upper bounds, we are guaranteed that a well-generalized model can be obtained even when the amount of data is limited. (2) Providing a regularization-based learning method for DG when facing limited data. This method not only underscores the importance of the number of domains $E$ for stability and generalization error bounds but also exhibits Q-linear convergence.

## 2 METHODOLOGIES

In this section, we first introduce the notations used as the basis of our derivation. We then provide a general exponential-decay upper bound based on the empirical and expected errors of trained models on data where the distributions are not independent and identically distributed. Using this bound, we derive the stability bound for models trained with regularization-based methods under the assumption that the hypothesis space is a reproducing kernel Hilbert space. Consequently, we analyze the stability and generalization error bounds of regularized mean squared error in the binary classification setting. NOTE that this generalization error bound clearly exhibits how we address the less-sample problem in domain generalization. Furthermore, we present a general learning objective for applying our theoretical results. In a similar manner, we analyze the stability, generalization bounds, and convergence of the proposed method. Finally, we discuss the limitations of our approach.

### 2.1 PRELIMINARIES

We here introduce some notation and related concepts that will be used to derive our methods. Considering $\mathcal{X} \subset \mathbb{R}$ and $\mathcal{Y} \subset \mathbb{R}$ being respectively an input and output space, we have a training set

$$D = \{Z_1 = (X, Y)_1, \ldots, Z_E = (X, Y)_E\}$$

of size $E$ in $\mathcal{Z} = \mathcal{X} \times \mathcal{Y}$ drawn from an unknown distribution $\mathcal{P}$. Assume that each $Z \in D$ contains $N$ data pairs $(x_j, y_j)$, i.e., $(X, Y) := \{(x_j, y_j)\}_{j=1}^N$. Under domain generalization (DG), $\{(x_j, y_j)\}_{j=1}^N \sim^{i.i.d.} P_{Z^i}$ for all $i \in [E]$, and $P_{Z_i} \neq P_{Z_j}$ for all $i \neq j \in [E]$. This indicates that all data samples are ONLY independent but NOT identically distributed. We also consider a hypothesis $h \in \mathcal{H}$ as a deterministic mapping for prediction, i.e., $h : \mathcal{X} \to \mathcal{Y}$. Furthermore, we assume that all mappings are measurable and all datasets are countable, which does not restrict the generality of the results presented hereafter.

For consistency with the data settings in (Bousquet & Elisseeff, 2002), we also consider a modified training set obtained by replacing the $v$-th element in $D$ as follows:

$$D^v = \{Z_1, \ldots, Z_{v-1}, Z'_v, Z_{v+1} \ldots, Z_E\},$$

where the replacement sample $Z'_v$ is assumed to be drawn from $\mathcal{P}$ and is also independent of $D$, but not identically distributed with it. Note that $Z'_v$ also contains $N$ data pairs, which differs from the setting in (Bousquet & Elisseeff, 2002), where the replacement element is only a single data pair.

To measure the accuracy of predictions from $h$, we consider the cost function $c : \mathcal{Y} \times \mathcal{Y} \to \mathbb{R}_{\geq 0}$, and the loss of $h$ with respect to data pairs $(X, Y)$ is then defined as

$$\ell(h, Z) = \ell(h, (X, Y)) = c(h(X), Y).$$

Correspondingly, the expected risk of $h$ is defined as

$$R^{\exp}(h) = \frac{1}{E} \sum_{i=1}^{E} \mathbb{E}_{Z_i \sim P_{Z_i}}[\ell(h, Z_i)],$$

and the empirical risk of $h$ is defined as

$$R(h) = \frac{1}{E} \sum_{i=1}^{E} \ell(h, Z_i) = \frac{1}{EN} \sum_{i=1}^{E} \sum_{j=1}^{N} \ell(h, (x_j, y_j)_i).$$

Considering using different training sets, we have two types empirical risks, namely $R(h, D) = (1/E) \sum_{Z \in D} \ell(h, Z)$ and $R(h, D^v) = (1/E) \sum_{Z \in D^v} \ell(h, Z)$. In the following, we will also use the shorthand notations $R(h) \equiv R(h, D)$ and $R'(h) \equiv R(h, D^v)$. Moreover, there is a concept that we hope the readers keep in mind: $\ell(h, Z) = (1/N) \sum_{z \in Z} \ell(h, z)$.

## 2.2 THEORETICAL BOUNDS

The stability of hypotheses is important for deriving the goal: we aim to obtain bounds on the generalization error for hypotheses and want these bounds to be tight when the hypotheses satisfy the stability conditions. Bousquet & Elisseeff (2002) provided many ways to define the stability of the outputs of learning algorithms. Motivated by their work, we provide the following definition:

**Definition 1** (Error Stability). *A hypothesis $h \in \mathcal{H}$ has error stability $\beta$ with respect to the empirical risk if the following holds*

$$\forall D \in \mathcal{Z}^E, \forall v \in [E], |R(h, D) - R(h, D^v)| \leq \beta, \tag{1}$$

*which can be also be written*

$$\forall D \in \mathcal{Z}^E, \forall v \in [E], \frac{1}{EN} \left| \sum_{Z \in D} \sum_{z \in Z} \ell(h, z) - \sum_{Z' \in D^v} \sum_{z' \in Z'} \ell(h, z') \right| \leq \beta. \tag{2}$$

**Remarks**: (a) Hereafter, we briefly consider that $|R(h) - R'(h)| \leq \beta$. (b) Inequality (1) denotes the risk difference for a single $h$ under different datasets $D$ and $D^v$, which differs from previous works that consider the risk difference for different hypotheses $h$ and $h'$ trained on different datasets $D$ and $D^v$. (c) We consider the empirical risk, which differs from the previous case, where the expected risk was considered. (d) In simple terms, we hope that the empirical risk difference for any $h$ across different datasets is smaller than a positive constant $\beta$. If this $\beta$ is infinitesimally small, it implies that $h$ will perform well on any dataset related to the training data. This is indeed the goal of domain generalization.

Based on the above definition of error stability, we can directly apply McDiarmid's inequality to obtain the following exponential decay bound:

**Theorem 1.** *For any measurable function $R : \mathcal{Z}^E \to \mathbb{R}_{\geq 0}$, any hypothesis $h \in \mathcal{H}$ that satisfies the stability condition in Definition 1, i.e., $h$ has $\beta$-stability, and any $\varepsilon > 0$, the following inequality holds*

$$P\left[R(h, D) - R^{\exp}(h, D) \geq \varepsilon\right] \leq \exp\left\{\frac{-2\varepsilon^2}{E\beta}\right\}, \forall D \in \mathcal{Z}^E. \tag{3}$$

**Remarks**: (a) We have $\varepsilon = \sqrt{((E\beta)/2) \ln(1/\delta)}$ and then, with probability at least $1-\delta$, $R-R^{\exp} \leq \sqrt{((E\beta)/2) \ln(1/\delta)}$. (b) As expected, a more restrictive stability criterion yields a correspondingly tighter bound. *All proofs of the theoretical results are provided in Appendix A.3.*

Based on the above result, we can see that the key is to find the $\beta$, as a small $\beta$ leads to a small generalization error. In the following, we mainly focus on deriving this $\beta$. We begin by considering the $\sigma$-admissibility of loss functions for analyzing the difference $|R(h) - R'(h)|$.

**Definition 2.** *A loss function $\ell$ defined on $\mathcal{H} \times \mathcal{Y}$ is $\sigma$-admissible with respect to $\mathcal{H}$ is the associated cost function $c$ is convex and the following condition holds*

$$\forall h, h' \in \mathcal{H}, \forall(X, Y) \in \mathcal{Z}, |\ell(h, (X, Y)) - \ell(h', (X, Y))| \leq \sigma |h(X) - h'(X)|.$$

**Remarks**: (a) The $\sigma$-admissibility only specifies a restricted negative value to guarantee continuity or stability, but not boundedness. (b) Recall the definition of the loss function: the above means that $|c(h(X), Y) - c(h'(X), Y)| \leq \sigma |h(X) - h'(X)|$. (c) The above inequality denotes that the loss difference for two hypotheses cannot be larger than $\sigma$ times the prediction difference.

We then define $\Delta h = |h(X) - h'(X)|, \forall h, h' \in \mathcal{H}$. We assume that the functional $f(h) = |R(h) - R'(h)|$ has a minimum (not necessarily unique) in $\mathcal{H}$, and let $h^*$ denote a minimizer, i.e., $h^* = \arg\min_{h' \in \mathcal{H}} |R(h') - R'(h')|$. We assume that $|R(h^*) - R'(h^*)| \leq \beta^*$, which implies that the minimizer $h^*$ achieves $\beta^*$-stability. Here, we should remind the readers that $h^*$ is the minimum only for the functional $f(h)$, rather than for $R(h)$ or $R'(h)$. In fact, this assumption is less restrictive than assuming that $R(h)$ and $R'(h)$ attain minimum points, since the data are not i.i.d. Consequently, $R(h^*)$ or $R'(h^*)$ may not be minimal at $h^*$. If we forcefully assume that $R(h)$ and $R'(h)$ attain their minimum points, we cannot guarantee that this will lead to the minimum of $f(h)$. Therefore, we only consider that $f(h)$ attains its minimum.

The functional $f$ represents the difference in empirical risk of a given hypothesis $h$ across two datasets. Now, we may relate it to $\sigma$-admissibility of the loss function. Accordingly, we present the following inequality

$$|R(h) - R'(h)| \leq \sigma(\Delta h_D + \Delta h_{D^v}) + \beta^*, \tag{4}$$

where $\Delta h_D$ and $\Delta h_{D^v}$ are defined respectively as $|h(X) - h^*(X)|_D$ and $|h(X) - h^*(X)|_{D^v}$. The subscripts $D$ and $D^v$ denote the sets over which the values of $X$ are taken. The detailed derivation of Inequality (4) is provided in Appendix A.3. Here, we obtain that the empirical risk difference for an $h$ is less than some quantity; then, we consider the risks of hypotheses obtained by regularization-based learning objectives to further explore $\beta$. The learning objective is $\min_{h \in \mathcal{H}} R_r(h, D, \lambda)$, where $h$ is parametrized and

$$R_r(h, D, \lambda) = \frac{1}{E} \sum_{i=1}^{E} \ell(h, Z_i) + \lambda N(h),$$

Here, $\lambda$ is a penalty factor, and $N(h)$ represents any regularization for $h$. This learning method follows the Structural Risk Minimization (SRM) principle, as $N(h)$ is typically used to control the complexity (VC-dimension) of $h$. When using the dataset $D^v$, we obtain $R'_r(h, D^v, \lambda)$. Similarly, we denote by $R_r(h)$ and $R'_r(h)$ the corresponding shorthand notations.

Then, let us present a general result with reference to $R_r(h)$ and $R'_r(h)$, or, more specifically, to $N(\cdot)$, which will lead to the derivation of stability bounds.

**Lemma 1.** *Let $\ell$ be $\sigma$-admissible with respect to $\mathcal{H}$, and let $N$ be a functional defined on $\mathcal{H}$. Assume that $h^* = \arg\min_{h' \in \mathcal{H}} |R(h') - R'(h')|$ and $\nabla R(h^*) \geq 0$. For any $t \in [0, 1]$, $R_r(h^*)$, and $R'_r(h^*)$, we have*

$$N(h^*) - N(h^* + t\Delta h) \leq \frac{\sigma t}{2\lambda EN} (\Delta h_D + \Delta h_{D^v}). \tag{5}$$

**Remarks**: (a) This result leads to the regularizations associated with the $\sigma$-admissibility of loss functions. (b) This result is more general than Lemma 20 in (Bousquet & Elisseeff, 2002), since Lemma 20 requires the existence of the minima of $R_r(h)$ and $R'_r(h)$, whereas we only assume that the function $f(h)$ attains its minimum. (c) Importantly, this result allows us to consider any $h$ obtained by the learning objective, instead of only the minimum (if it exists), thereby ensuring that the following derivations apply to any $h$.

Next, we impose a specific assumption on $\mathcal{H}$ to ensure the well-defined formulation of $N(\cdot)$. In particular, we consider regularization within a reproducing kernel Hilbert space (RKHS). The choice of an RKHS is made to enable efficient computation and generalization of kernel methods, while ensuring that the function space is complete, stable, and well-controlled. Importantly, Bousquet & Elisseeff (2002) established several lemmas that can be directly utilized in our derivation within the RKHS framework. Then, we consider that

$$N(h) = \|h\|_k^2,$$

where $k$ refers to the kernel. Typically, we state that $\mathcal{H}_k$ denotes the norm in an RKHS, but for simplicity, we use $k$ here. Based on the fundamental property (reproducing) of an RKHS $\mathcal{H}$, we have

$$\forall h \in \mathcal{H}, \forall X \in \mathcal{X}, h(X) = \langle h, k(X, \cdot) \rangle.$$

Then, according to Cauchy-Schwarz inequality, we have

$$\forall h \in \mathcal{H}, \forall X \in \mathcal{X}, |h(X)| \leq \|h\|_k \sqrt{k(X,X)}. \tag{6}$$

We now present the error stability of regularized learning methods in an RKHS, stated as the following result.

**Theorem 2.** *Assume that $\mathcal{H}$ is a reproducing kernel Hilbert space with kernel $k$ such that $\forall X \in \mathcal{X}, \sqrt{k(X,X)} \leq \kappa \leq \infty$. Let $\sigma$-admissible with respect to $\mathcal{H}$. Any hypothesis $h$ achieved by*

$$\min_{h \in \mathcal{H}} \frac{1}{E} \sum_{i=1}^{E} \ell(h, Z_i) + \lambda \|h\|_k^2, \tag{7}$$

*has error stability $\beta$ respect to $\ell$ with*

$$\beta \leq \frac{2\sigma^2 \kappa^2}{\lambda E N} + \beta^*.$$

**Remarks**: (a) The $\beta$ is inversely proportional to the product of the penalty factor and the total number of data samples. (b) There is a fixed quantity $\beta^*$, and $\beta$ is determined solely by the first item when $\beta^*$ is infinitesimal. Since $\beta^*$ represents the optimal stability, it can be treated as a constant approaching zero when $h^*$ exists. However, we cannot omit it, as we are constrained by the data conditions in domain generalization.

Here, we have found the desired $\beta$. We then consider a specific example illustrating the application of the above results: Regularized Least Squares Regression (RLSR). This indicates that we consider the loss function $\ell(h, Z) = (h(X) - Y)^2$ in Formula (7). Other applications, such as classification or those using different loss functions or tasks, can also follow the following process. Note that the $\sigma$-admissibility cannot provide a bound; you need to invoke Lemma 5 to complete the argument.

**Corollary 1** (Stability and Generalization bound for RLSR)**.** *Consider $\mathcal{Y} = [0, B]$ and the loss function $\ell(h, Z) = (h(X) - Y)^2$. We state that $\ell$ is $2B$-admissible. Also, we have*

$$\forall Z \in \mathcal{Z}, 0 \leq \ell(h, Z) \leq \kappa \sqrt{\frac{B}{\lambda}}.$$

*The stability bound for any $h \in \mathcal{H}$ obtained by RLSR is*

$$\beta \leq \frac{8B^2 \kappa^2}{\lambda E N} + \beta^*,$$

*and, with probability at least $1 - \delta$, we have the generalization error bound*

$$R(h) - R^{\exp}(h) \leq 2B\kappa \sqrt{\frac{\ln(1/\delta)}{\lambda N}} + \sqrt{\frac{E\beta^* \ln(1/\delta)}{2}}. \tag{8}$$

**Remarks**: (a) This bound (8) indicates how the regularization-based learning methods address the small-sample problem. When $N$ is small, we need to increase $\lambda$. (b) There is an advantage compared to the bounds based on VC-dimension, namely that it explicitly expresses the relationship between learning objectives and the generalization error of trained models through a simple $\lambda$. (c) For a small fixed item $\sqrt{(E/2)\beta^* \ln(1/\delta)}$, we obtain an order of $\mathcal{O}(1/\sqrt{\lambda N})$.

Finally, we discuss the choice of the kernel in RKHS, which determines the magnitude of $\kappa$. Assuming a linear kernel and that normalization techniques are applied to $X$, we have $\kappa = 1$. This emphasizes the importance of normalizing the input data. For the Gaussian kernel, $\kappa$ is also 1. Therefore, the choice of kernel can be made according to specific considerations.

## 2.3 LEARNING METHODS

In the previous subsection, we analyzed the stability and generalization error bounds for regularization-based learning methods. We also knew how to address the less-sample problem in domain generalization. Here, we provide a general learning objective to apply the previous results:

$$\min_h R(h) \quad s.t. \sum_{i \in [E]} \|h\|_2^2 \leq \eta, \eta > 0, \tag{9}$$

where $R(h)$ on $D$ is defined above and $h$ is parametrized. We then consider its regularization through the Lagrangian function

$$L(\lambda, h) = R(h) + \lambda \sum\nolimits_{i \in [E]} \|h\|_2^2,$$

as follows:

$$\min_h R(h) + \lambda \sum_{i \in [E]} \|h\|_2^2. \tag{10}$$

Before analyzing using the results in the previous section, we should make some illustrations for the above learning objective. This learning objective, which we call DLAERM, is essentially a relaxed version of ERM subject to the $L_1$ norm (which we call LASERM). This is because LASERM requires global domain sparsity, whereas DLAERM only requires inter-domain sparsity. The reason for the inter-domain sparsity is that $\sum_{i \in [E]} (\|h\|_2)^i = (\|h\|_2)^1 + (\|h\|_2)^2 + \cdots + (\|h\|_2)^E$ is essentially $L_1$ norm. *If this explanation is insufficient for understanding, please consider the relationship between LASSO and group LASSO.* This inter-domain sparsity requires that $E \geq 2$. When $E = 1$, Formula (10) reduces to the ridge regression problem under the mean squared error (MSE) as the empirical risk. Besides the motivation provided by Formula (7), considering inter-domain sparsity and inner smoothness is another motivation for the proposed DLAERM.

Then, we use the results from the previous subsection to analyze the stability bound and the generalization error of Formula (10). For this application, the space $\mathcal{H}$ is simply considered a finite-dimensional Euclidean space, which represents a simple case of an RKHS. Here, we have $N(h) = \sum_{i \in [E]} \|h\|_{2,k}^2$. Thus, based on Lemma 1, Theorem 1, and the basic setting in Corollary 1, we have the stability bound for any $h \in \mathcal{H}$ obtained by Formula (10) as

$$\beta \leq \frac{8B^2\kappa^2}{\lambda E^2 N} + \beta^*, \tag{11}$$

and, with probability at least $1 - \delta$, the generalization bound is

$$R(h) - R^{\exp}(h) \leq 2B\kappa\sqrt{\frac{\ln(1/\delta)}{\lambda EN}} + \sqrt{\frac{E\beta^* \ln(1/\delta)}{2}}. \tag{12}$$

Considering a linear kernel and then applying normalization to $X$, we have $\kappa = 1$. Here, we need to explain why $E^2$ appears in the stability bound above. This is because $N(h) = \sum \|h\|_{2,k}^2$, and, based on Lemma 1, we obtain a $\beta$ that depends on the square of the number of domains. This provides a significant advantage for our learning method: according to the bound (12), we obtain a faster rate $1/\sqrt{\lambda EN}$ than that of the bound (8). Moreover, the stability bound above is more compact than the stability bound for Inequality (8). Consequently, as long as $E > 1$, we have a guarantee that stable prediction models can be learned more quickly, even when the number of domains is small.

Next, we aim to analyze DLAERM (10) from the perspective of optimization theory with respect to its convergence. First, we state the following assumption:

**Assumption 1** (gradient descent method)**.** *Assume that the learning objective function $L(\theta)$ is smooth, and that the parameters $\theta$ are optimized using the gradient descent method, which is defined as*

$$\theta^{k+1} = \theta^k - \alpha_k \nabla L(\theta^k), \tag{13}$$

*where $k$ denotes the epoch index and $\alpha$ is the learning rate.*

We use this assumption to align with the optimization methods applied to model parameters in most deep learning methods, namely the gradient descent method. Note that we assume all learning objective functions are smooth. Consequently, we provide the following theorem for the convergence of DLAERM.

**Theorem 3.** *For the learning objective function (10), which is $\mu$-strongly convex, if the learning rate $\alpha_k$ is constant and satisfies $0 < \alpha \leq 1/(\mu + \mathbf{L})$, where $\mathbf{L}$ is the Lipschitz constant of the gradient, then the sequence $\{h^k\}$ generated by the iterative scheme (13) converges to $h^*$. Moreover, the convergence is Q-linear.*

We have now completed all analyses of our method.

## 2.4 DISCUSSION

In this subsection, we discuss several aspects of our work, including the differences from the work by Bousquet & Elisseeff (2002), other classical bounds for generalization errors, and the limitations of our method, among others.

**Differences**: Here, we only discuss the key differences, namely the assumption on data distribution and the assumption on minimum points. In their work, the data are i.i.d., whereas our assumption only requires the data distributions to be independent. Regarding the minimum points, their work assumes the existence of minima for both $R_r(h)$ and $R'_r(h)$, whereas we only assume the existence of a minimum $h^*$ for the functional $f(h)$. Consequently, their bounds are typically related to the minimum points, i.e., $\eta = \arg\min_{h \in \mathcal{H}} \sum \ell(h, Z_i) + \lambda \|h\|_k^2$. In contrast, our bounds apply to any learned $h$ through SRM, which may not necessarily be a minimum.

**Other bounds**: Here, we briefly discuss some bounds, but the specific details are omitted. For further information, please refer to the references. The classical VC-bound (Vapnik, 2013) and the bounds in (Bousquet & Elisseeff, 2002) are not included due to their different data assumptions. One well-known work on distribution-shifted data is presented by Ben-David et al. (2010), whose bounds also indicate that reducing model complexity can lead to reduced generalization error. However, their bounds heavily depend on the computation of the $\mathcal{H}\Delta\mathcal{H}$-divergence, which is difficult to estimate in real applications. Tong et al. (2023) presented an upper bound for their method, but the connection between the hyperparameters and the bound was not built. Recently, a lower bound for domain generalization was presented by Wang et al. (2024) to explain a phenomenon by Gulrajani & Lopez-Paz (2020), but it required an infinite number of data samples in each domain.

**Limitations of Ours**: (a) Unclear assumption regarding labeling noise. Although we mentioned that labeling noise should be considered when using data augmentation techniques, our theoretical results do not include a corresponding noise assumption, such as the Massart noise condition (Massart & Nédélec, 2006). As a result, the noise condition in our assumed data is unclear, and for data with high noise conditions, our method may not serve as an effective guidance. (b) Existing a constant $\beta^*$. If this $\beta^*$ is large, it will cause our bound to become very loose. This scenario also appears in (Bousquet & Elisseeff, 2002), for example, the $\lambda$ in their bound. However, we consider this a tradeoff condition, which represents a hard case for learning across different domains and is natural. This tradeoff implies that the trained models may not be optimal for each domain individually but perform well across all domains. Hence, we argue that this limitation is inevitable. (c) Limitations of the assumption on the hypothesis space. We only consider the RKHS. For other spaces where pointwise evaluation may be undefined or discontinuous, our bound cannot be established. For example, in the $L^2$ space, the Dirac delta function does not belong to $L^2$, and functions in $L^2$ are not necessarily continuous.

**Others**: Here, we briefly discuss the limitations of using data augmentation techniques and the synthetic data generated by large models. One main drawback of data augmentation techniques is that they lack a theoretical guarantee for maintaining unchanged labeling noise conditions after increasing the data size. We discussed this issue in the Introduction; thus, we do not elaborate further here. We instead focus on synthetic data generated by large models. We adopt the view proposed by Shumailov et al. (2024), which considers that using synthetic data may lead to model collapse. Consequently, we cannot fully trust the data generated by large models. However, some researchers argue that these methods can effectively address the less-sample problem. Therefore, the issue remains under debate.

## 3 EXPERIMENTS

In this section, we first present the validation results for the stability bound (11), i.e., $\beta \leq C/(\lambda E^2 N) + \beta^*$ using synthetic data, which are similar to those in Assumption 1 of (Peters et al., 2016) or the data defined in Formulas (1) – (3) of (Rosenfeld et al., 2020). Since we adopt synthetic data, the best stability $\beta^*$ is considered to be 0. Thus, we only observe $\beta \leq C/(\lambda E^2 N)$, and according to this result, we obtain a bound for $\lambda$:

$$\lambda \leq \frac{C}{\beta E^2 N}.$$

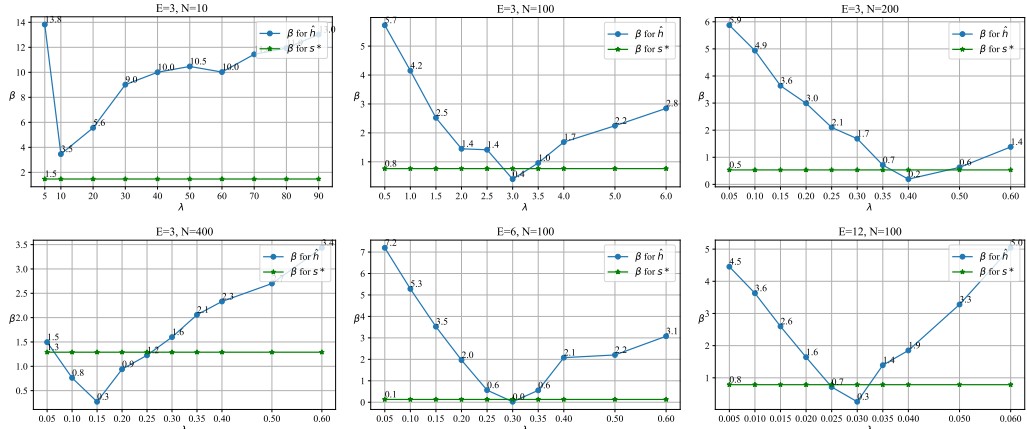

Figure 1: Validations on Stability bound (11). The x-axis denotes the values of $\lambda$, and the y-axis represents the values of $\beta$. $\hat{h}$ is learned by DLAERM on the dataset $D$, and $s^*$ is the optimal model. As observed, when $EN$ is small, the values of $\lambda$ are correspondingly large. Moreover, these results exhibit the tradeoff point for $\lambda$, which is the minimal $\beta$ point for $\hat{h}$.

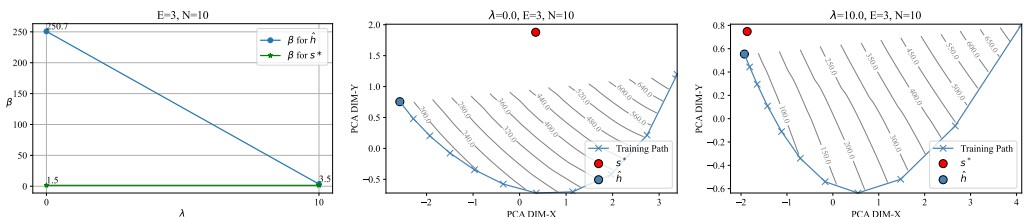

Figure 2: Comparisons of the stability of learned models by ERM and DLAERM, as well as the learning convergence analyses. The first subfigure on the left shows the stability results, and the remaining subfigures show the results for the learned models. Here, $\lambda = 0$ in each subfigure denotes the results of ERM, and $\lambda = 10$ denotes the results of DLAERM. The representations of $s^*$ and $\hat{h}$ are expressed by the two-dimensional PCA projections shown as red and blue points. The training path represents the learned $h$ across multiple epochs, and the gray lines indicate specific validation loss values on $D^v$. As observed, with a large value of $\lambda$, $\hat{h}$ approaches $s^*$ and achieves a lower value of $\beta$. This clearly illustrates how regularization-based methods address the less-sample problem.

This indicates that if $\beta \to \beta^* = 0$, the bound increases and $\lambda$ can be arbitrarily large. However, since we know that if $\lambda$ is too large, it reduces the freedom of learned models in regularization-based methods, $\beta$ will alternatively increase when $\lambda$ becomes too large, which causes the bound on $\lambda$ to decrease. This suggests that there is a tradeoff point for $\lambda$: when the value of $\lambda$ surpasses this tradeoff point, further increasing $\lambda$ will lead to poor stability. Therefore, this tradeoff phenomenon should be observed in the validation results. Finally, we discuss how regularization-based methods address the limited-sample problem by using the learning convergence analyses. The experiments on real-world datasets for image classification are provided in Appendix A.5.

### 3.1 BASAL SETTINGS

**Data**: The specific data structure of $(x, y)$ in the synthetic data is as follows, and this structure is commonly used in domain generalization studies:

$$y = \omega z_c + \varepsilon, \quad z_c \perp \varepsilon, \quad z_e = y + \mathcal{N}(0, e), \quad x = [z_c, z_e],$$

where $z_c \leftarrow \mathcal{N}(0, 1)$, $\varepsilon \leftarrow \mathcal{N}(0, 1)$, $z_c, z_e \in \mathbb{R}^d$, $x \in \mathbb{R}^{2d}$, $\omega \in \mathbb{R}^d$, and $y \in \mathbb{R}$. Here, the $e$ represents the domain intervention on the variance of the Gaussian distributions, and we set $d = 20$. By setting different values of $e$, we can obtain an $E$ domain dataset $D$. For each domain $i \in [E]$, we generate $N$ pairs of $(x, y)$ and obtain $(X, Y)_i$. Note that, for simplicity, we keep $s^*$ fixed across

domains, i.e., $s^* = [\omega, \mathbf{0}] \in \mathbb{R}^{2d}$, where $\omega \leftarrow \mathcal{N}(0, 1)$. Note that $s^*$ is not equivalent to $h^*$ that leads to the best stability $\beta^*$. If one aims to observe $\beta^*$, set $\varepsilon = 0$, and then the $\beta$ computed by $s^*$ is $\beta^*$. The $E$ domains in the training set $D$ are defined as $\{e_i = i | i \in [E]\}$, whereas, for the validation set $D^v$, we replace the last domain in $D$ with $e = E + 1$, i.e., the $E$ domain in the validation set $D^v$ are defined as $\{e_i = i | i \in \{1, \ldots, E - 1, E + 1\}\}$. For example, if we consider 3 domains of $D$ and 3 domain of $D^v$, these domains are $\{1.0, 2.0, 3.0\}$ and $\{1.0, 2.0, 4.0\}$, respectively. Then, by setting different values for $E$ and $N$, we can obtain different $D$ and $D^v$. Note that, since our main focus is on the less-sample problem, $E$ and $N$ will not be particularly large.

**Others**: Based on the dataset $D$, we aim to learn a regression model $\hat{h}$ using Formula (10) with the loss function $\ell(h, Z) = (h(X) - Y)^2$. Then, according to Formula (1), we can compute the stability for $\hat{h}$ and $s^*$ using the datasets $D$ and $D^v$. Computing $\beta$ for $s^*$ aims to provide a reference for the $\beta$ of $\hat{h}$. Moreover, we can obtain the stability of the ERM models by setting $\lambda = 0$.

### 3.2 RESULTS

The main validation results on the stability bound for models learned by DLAERM are shown in Figure 1, and the corresponding learning convergence analyses for the models with minimal $\beta$ are presented in Figure 3. Moreover, we report the stability bounds for models learned by ERM (setting $\lambda = 0$ in DLAERM) and DLAERM, along with the convergence analyses, in Figure 2.

**Conclusions**: (a) The inversely proportional relationship between $\lambda$ and $EN$ is established, meaning that a larger $EN$ corresponds to a lower $\lambda$, and vice versa. As observed in Figure 1, the values of $\lambda$ are relatively small when $EN$ is large. (b) The optimal stability leads to good convergence. As observed in Figure 3, the learned models that achieved the minimal $\beta$ in Figure 1 are generally close to the preset true model. (c) The regularization-based learning methods can effectively address the less-sample problem in domain generalization. As observed in Figure 2, the stability of the ERM model is very high, whereas the stability of the DLAERM model is low under the same data conditions. Moreover, based on the convergence analyses, the DLAERM model is closer to the preset true model.

**Others**: (a) A tradeoff point for $\lambda$ is observed. Through Figure 1, we can clearly see the $V$ shape in the results of $\beta$ for $\hat{h}$. Once that tradeoff point is surpassed, increasing $\lambda$ leads to a larger $\beta$. (b) The squared $E$ is validated. When $E = 3$ and $N = 200$ or $N = 400$, and when $E = 6$ and $N = 100$, the values of $\lambda$ are similar, whereas when $E = 12$ and $N = 100$, the values of $\lambda$ are smaller than those when $E = 3$ and $N = 400$, despite the total number being the same. (c) $s^*$ indeed differs to $h^*$. We observe that $\beta$ for $s^*$ is typically smaller than $\beta$ for $\hat{h}$. However, when $\hat{h}$ achieves the optimal stability, $\beta$ for $\hat{h}$ is small than that of $s^*$. Sometimes, $\beta$ for $\hat{h}$ reaches $\beta^*$, as illustrated in Figure 1 when $E = 6$ and $N = 100$. This result strengthens the advantage of our assumption that the existing $h^*$ leads to the best stability, as the minimizer of the learning objective functions, i.e., $s^*$ in our experiments, cannot always lead to the best stability.

## 4 CONCLUSION

In this study, we present stability bounds for regularization-based learning methods. This allows us to use these learning methods under domain-shifted data, thereby addressing the less-sample problem. Importantly, we provide guidance on how to derive the upper bounds for different cost functions, making them applicable to real-world data. We have already discussed the limitations of our theoretical results in Section 2.4. Here, we discuss some thoughts on our learning method, which are raised by the tradeoff point for $\lambda$.

When the data samples are limited, based on our theoretical results, we can set a larger value for $\lambda$. However, due to the tradeoff point, this value for $\lambda$ may not be larger than $1.0$. This seems to indicate that the role of $\lambda$ is to act as a selector for data. For example, if we set $\lambda = 0.5$ and have $EN = 100$, based on our stability bounds, only 50 data samples are effective for learning. Moreover, for our learning method, $\lambda$ also serves as a selector of weights. This selector action, implemented via setting $\lambda$, shows that learning well-generalized models may not require too many data samples, which does not align with the large-number rule in learning theory. Addressing this concern will guide our future work.

REPRODUCIBILITY STATEMENT

We state that our theoretical and experimental results are both reproducible. First, for the reproducibility of theoretical results, we provide the specific proofs in Appendix A.3, and for some unproved lemmas, we cite the original works. Then, for experimental results, we provide detailed settings for the experiments in Appendix A.5 and the learning algorithms. However, the code is omitted, as the learning methods we used were provided by previous works, for which we include weblinks to download the code.

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

## A APPENDIX

In this appendix, we present the related works, lemmas, proofs, and several experimental results.

### A.1 RELATED WORKS

We briefly introduce some learning methods in domain generalization and then present some theoretical bounds. For more details, see the references.

**Learning Methods**: Since IRM (Arjovsky et al., 2019) was proposed, several studies have followed its primary idea and introduced further innovative theories and methods, such as IRM-Games (Ahuja et al., 2020), the invariant information bottleneck (Li et al., 2022), P-IRM (Choraria et al., 2023), etc. Moreover, there are several methods that aim to discover invariant representations in the latent space, such as (Muandet et al., 2013; Li et al., 2018). These methods assume that learning invariant features from available domain data can lead to well-generalized reasoning models on Out-of-distribution domains. However, several studies have questioned the effectiveness of Invariance-based methods, systematically analyzed their failure conditions (Rosenfeld et al., 2020), and proposed low-risk optimization problems (Wang et al., 2022). Interestingly, some researchers have introduced causal mechanisms to address the domain shift problem, such as (Christiansen et al., 2021; Wang et al., 2023). The causality-based methods, however, require more constraints than those invariance-based methods; thus, we question their performance under real-world data conditions. The discussion of using regularization in machine learning (ML) has recently attracted growing interest, as in (Xu et al., 2024; Zhang et al., 2025; Noori Saray et al., 2025; Li et al., 2023). Moreover, several theoretical studies discuss the complexity of neural networks, as in (Galanti et al., 2023; Levy & Abramovich, 2023).

**Theoretical Bounds**: The theoretical bounds in learning theory provide a guide for how much better the estimator is. From the VC bounds (Vapnik & Chervonenkis, 2015; Vapnik, 2013) to the Massart-noisy bounds (Massart & Nédélec, 2006), the mystery of the learning pattern in machine learning has been gradually revealed. These methods provide several useful fundamental inequalities to analyze different problems, inspiring future studies, including ours. Moreover, Bousquet & Elisseeff (2002) present the concept of stability for learning methods via concentration inequalities. However, these bounds require that the data be i.i.d., which contradicts the data assumptions in domain generalization. Study (Ben-David et al., 2010) was the first to focus on learning from different domains, providing generalization bounds based on the VC-class hypothesis and the $\mathcal{H}$-divergence, which highlights their limitation in guiding the learning. Wang et al. (2024) recently presented a lower bound under the VC-class and Massart-noisy condition, but their bound requires an infinite number of data samples in each domain. We consider this a huge drawback for guiding the learning.

### A.2 LEMMAS

**Lemma 2.** $F = \sum_i^n \alpha_i f_i$, where $\alpha_i > 0, n \geq 1$, represents a (strongly) convex function if each $f_i$ is (strongly) convex.

**Lemma 3** (Convergence of the gradient descent method for strong convex functions). *Assume $L(\theta)$ is $\mu$-strongly convex and $\mathbf{L}$-Lipschitz gradient continuous function, and that $L^* = L(\theta^*) = \inf_\theta L(\theta)$ exists and is attainable. If the learning rate $\alpha_k$ is constant and satisfies $0 < \alpha \leq 1/(m + \mathbf{L})$, then the sequence $\{\theta^k\}$ of function values generated by the gradient descent scheme converges to the optimal value. Moreover, the convergence rate is $\mathcal{O}(c^k)$ in terms of the function values.*

**Lemma 4** (McDiarmid's inequality (McDiarmid et al., 1989)). *Let $D$ and $D^v$ defined as in Section 2.1, Let $R : \mathcal{Z}^E \to \mathbb{R}_{\geq 0}$ be any measurable function for which there exists constants $c_v(v = 1, \ldots, E)$ such that*

$$\sup_{Z'_v \in \mathcal{Z}} |R(D) - R(D^v)| \leq c_v,$$

*then*

$$P[R(D) - \mathbb{E}_D[R(D)] \geq \varepsilon] \leq \exp\left\{\frac{-2\varepsilon^2}{\sum_{v \in [E]} c_v}\right\}.$$

**Lemma 5.** *Let $h$ be the hypothesis obtained by Formula* (7) *where $\ell$ is a loss function associated with a convex cost function $c(\cdot, \cdot)$. We denote by $B(\cdot)$ a positive non-decreasing real-valued function such that for all $y \in \mathcal{Y}$. Then, we have*

$$\forall y' \in \mathcal{Y}, c(y, y') \leq B(y)$$

*For any training set $D$, we also have*

$$\|h\|_k^2 \leq \frac{B(0)}{\lambda}$$

$$\forall z \in \mathcal{Z}, 0 \leq \ell(h, Z) \leq B\left(\kappa\sqrt{\frac{B(0)}{\lambda}}\right).$$

*Moreover, $\ell$ is $\sigma$-admissible where $\sigma$ can be taken as*

$$\sigma = \sup_{y' \in \mathcal{Y}} \sup_{|y| \leq B\left(\kappa\sqrt{\frac{B(0)}{\lambda}}\right)} \left|\frac{\partial c}{\partial y}(y, y')\right|.$$

### A.3 PROOFS

We first present the proofs of the theorems in the main paper, and then provide the proofs of the lemmas used.

*Proof of Theorem 1.* Here, we replace $R(D)$ with $R(h, D)$ and $\mathbb{E}_D[R(D)]$ with $R^{exp}(h, D)$ to consider the hypotheses, and $R(h, D)$ and $R^{exp}(h, D)$ still represent random variables, which still satisfy the requirements of McDiarmid's inequality.

Since we consider that any $h \in \mathcal{H}$ satisfies $\beta$-stability, we have

$$\forall h \in \mathcal{H}, \forall v \in [E], \forall Z'_v \in \mathcal{Z}, |R(h, D) - R(h, D^v)| \leq \beta.$$

This indicates that

$$\sum_{v \in [E]} \sup_{Z'_v \in \mathcal{Z}} |R(h, D) - R(h, D^v)| \leq E\beta, \forall h \in \mathcal{H}.$$

Then, by applying Lemma 4, the proof is completed. $\qquad\square$

*Derivation Process of Inequality* (4). Since, we have $h^* = \arg\min_{h' \in \mathcal{H}} |R(h') - R'(h')|$. Then, for any $h \in \mathcal{H}$, we have

$$\begin{aligned}
|R(h) - R'(h)| &= |R(h) + R(h^*) - R(h^*) + R'(h^*) - R'(h^*) + R'(h)| \\
&\leq |R(h) - R(h^*)| + |R'(h) - R'(h^*)| + |R(h^*) - R'(h^*)| \\
&\leq |R(h) - R(h^*)| + |R'(h) - R'(h^*)| + \beta^* \\
&\leq \sigma|h(X) - h^*(X)|_D + \sigma|h(X) - h^*(X)|_{D^v} + \beta^*
\end{aligned}$$

The first inequality holds by the triangle inequality, the second holds under the assumption that $|R(h^*) - R'(h^*)| \leq \beta^*$, and the last holds due to the $\sigma$-admissibility of the loss functions. $\qquad\square$

*Proof of Theorem 2.* Since $N(h) = \|h\|_k^2$, based on Lemma 1, we have

$$t\|\Delta h\|_k^2 \leq \frac{\sigma t}{2\lambda EN}(\Delta h_D + \Delta h_{D^v}).$$

According to the assumption that $\forall X \in \mathcal{X}, \sqrt{k(X, X)} \leq \kappa \leq \infty$ and Inequality (6), we have

$$\begin{aligned}
\Delta h_D &\leq \|\Delta h\|_k\sqrt{k(X, X)} \leq \|\Delta h\|_k\kappa, \\
\Delta h_{D^v} &\leq \|\Delta h\|_k\kappa.
\end{aligned}$$

Consequently, we obtain

$$\|\Delta h\|_k^2 \leq \frac{\sigma}{2\lambda EN} 2\|\Delta h\|_k \kappa = \frac{\sigma\|\Delta h\|_k \kappa}{\lambda EN},$$
$$\|\Delta h\|_k \leq \frac{\sigma\kappa}{\lambda EN}$$

Moreover, according to Inequality (4), and by taking $\beta \leq \sigma(\Delta h_D + \Delta h_{D^v}) + \beta^*$, we have

$$\beta \leq 2\sigma\|\Delta h\|_k \kappa + \beta^*,$$

completing the proof.

$\square$

*Proof of Corollary 1.* Since we move Lemma 5 to the Appendix, we provide its proof here. By Lemma 5, the loss function is $2B$-admissible, and we obtain

$$\forall Z \in \mathcal{Z}, 0 \leq \ell(h, Z) \leq \kappa\sqrt{\frac{B}{\lambda}}.$$

Then, according to Theorem 2, we have

$$\beta \leq \frac{8B^2\kappa^2}{\lambda EN} + \beta^*.$$

At last, we apply Theorem 1 to obtain the final result, namely

$$\varepsilon = \sqrt{\frac{E\beta}{2}\ln(\frac{1}{\delta})} \leq \sqrt{\frac{E}{2}(\frac{8B^2\kappa^2}{\lambda EN} + \beta^*)\ln(\frac{1}{\delta})}$$
$$\leq \sqrt{\frac{4B^2\kappa^2\ln(1/\delta)}{\lambda N} + \frac{E\beta^*}{2}\ln(\frac{1}{\delta})}$$
$$\leq 2B\kappa\sqrt{\frac{\ln(1/\delta)}{\lambda N}} + \sqrt{\frac{E\beta^*}{2}\ln(\frac{1}{\delta})}.$$

$\square$

*Proof of Theorem 3.* First, we prove that Formula (10) is strongly convex and present its corresponding Lagrangian function as follows:

$$L(\lambda, \theta) = \sum_{e=1}^{E}\left\|\theta^T X^e - Y^e\right\|_2^2 + \lambda\sum_{e=1}^{E}\|h\|_2^2.$$

Without loss of generality, we consider the mean squares error as the loss function. Here, we use $\theta$ to replace $h$, since $h$ is parameterized, and use $e$ to denote the domain index.

Next, we prove that the function $l^e(\lambda, \theta) = \left\|\theta^T X^e - Y^e\right\|_2^2 + \lambda\|\theta\|_2^2$ under any environment $e$ is strongly convex. According to the definition of a $\mu$-strongly convex function, if

$$g(x) = f(x) - \frac{\mu}{2}\|x\|_2^2$$

is convex, then $f(x)$ is said to be $\mu$-strongly convex. Therefore, let us consider the function $f(t) = (tX - Y)^2 + \lambda\|t\|_2^2$. For notational simplicity, we omit the superscript $e$. Compute $f(t) - k\|t\|_2^2$, where $2k = \mu$. We have

$$g(t) = (tX - Y)^2 + \lambda\|t\|_2^2 - k\|t\|_2^2$$
$$= (tX - Y)^2 + (\lambda - k).$$

The function $g(t)$ is quadratic with an additive constant, and is therefore convex. As a result, $f(t)$ is $\mu$-strongly convex. Consequently, by Lemma 2, we know that $L(\lambda, \theta)$ is also strongly convex for all $e$. We then apply Lemma 3 to complete the proof.

$\square$

The following content presents proofs for certain lemmas; we only provide the proofs that have not appeared publicly or in others' works. For those lemmas that are well-known or have appeared in others' work, we provide citations and omit the proofs here.

*Proof of Lemma 1.* Since $h^* = \arg\min_{h' \in \mathcal{H}} |R(h') - R'(h')|$, for any $h \in \mathcal{H}$, we have

$$|R(h^*) - R'(h^*)| \leq |R(h^* + t\Delta h) - R'(h^* + t\Delta h)|.$$

Here, $h^* + t\Delta h \neq h^*$ for any $t > 0$. Then, we have

$$(R(h^*) - R'(h^*))^2 \leq (R(h^* + t\Delta h) - R'(h^* + t\Delta h))^2.$$

Let $\Phi(h) = |R(h) - R'(h)|$, we have $0 \in \nabla\Phi(h^*)$, which implies that $\exists g \in \nabla R(h^*) \cap \nabla R'(h^*)$. Note that here $0$ and $g$ belong to the hypothesis class rather than being scalars.

Now, let $\phi(t) = R(h^* + t\Delta h)$ and $\phi'(t) = R'(h^* + t\Delta h)$. Then, we have

$$\nabla_t \phi(t) = \nabla R(h^* + t\Delta h)\Delta h,$$
$$\nabla_t \phi'(t) = \nabla R'(h^* + t\Delta h)\Delta h.$$

Consider $\varphi(t) = \phi(t)\phi'(t)$, we have

$$\nabla_t \varphi(t) = \nabla R(h^* + t\Delta h)\Delta h\phi'(t) + \nabla R'(h^* + t\Delta h)\Delta h\phi(t),$$
$$\nabla_t \varphi(0) = \nabla R(h^*)\Delta h\phi'(0) + \nabla R'(h^*)\Delta h\phi(0).$$

Since $R$ is convex, its gradient is a monotone mapping, so we have $\nabla R(h^* + t\Delta h) \geq \nabla R(h^*)$ for any $t \geq 0$. Moreover, we assume that $\nabla R(h^*) \geq 0$. Then, we obtain $\nabla_t \varphi(t) \geq \nabla_t \varphi(0)$. This implies that

$$R(h^* + t\Delta h)R'(h^* + t\Delta h) \geq R(h^*)R'(h^*).$$

Then, we compute

$$(R(h^* + t\Delta h) + R'(h^* + t\Delta h))^2 - (R(h^*) + R'(h^*))^2$$
$$= (R_\Delta^2 + 2R_\Delta R'_\Delta + (R'_\Delta)^2) - (R^2 + 2RR' + (R')^2)$$
$$= (R_\Delta^2 + 2R_\Delta R'_\Delta + (R'_\Delta)^2 - 2R_\Delta R'_\Delta + 2R_\Delta R'_\Delta)$$
$$\quad - (R^2 + 2RR' + (R')^2 - 2RR' + 2RR')$$
$$= ((R_\Delta - R'_\Delta)^2 + 4R_\Delta R'_\Delta) - ((R - R')^2 + 4RR')$$
$$= ((R_\Delta - R'_\Delta)^2 - (R - R')^2) + 4(R_\Delta R'_\Delta - RR').$$

Based on the previous derivations, we obtain

$$(R(h^*) + R'(h^*))^2 \leq (R(h^* + t\Delta h) + R'(h^* + t\Delta h))^2$$
$$R(h^*) + R'(h^*) \leq R(h^* + t\Delta h) + R'(h^* + t\Delta h).$$

Then, we replace $R(h^*)$ and $R'(h^*)$ with $R_r(h^*)$ and $R'_r(h^*)$, respectively, and obtain

$$R_r(h^*) + R'_r(h^*) \leq R_r(h^* + t\Delta h) + R'_r(h^* + t\Delta h).$$

Consequently, we have

$$2\lambda(N(h^*) - N(h^* + t\Delta h)) \leq (R(h^* + t\Delta h) - R(h^*)) + (R'(h^* + t\Delta h) - R'(h^*))$$

Finally, based on the $\sigma$-admissibility of the loss functions and the definition of $R$, we obtain

$$N(h^*) - N(h^* + t\Delta h) \leq \frac{\sigma t}{2\lambda EN}(\Delta h_D + \Delta h_{D^v}).$$

$\square$

*Proof of Lemma 2.* We employ mathematical induction and assume $\exists \theta \in (0, 1)$, $\beta = 1 - \theta$, and $\forall (x, y) \in \mathbf{dom} f$. Based on the properties of (strongly) convex functions, we know that $\alpha f$ is a (strongly) convex function if $f$ is (strongly) convex and $\alpha > 0$. For convenience, we set $\alpha = 1$ for any $f$.

When $n = 1$ and $n = 2$, since $f_1 and f_2$ are convex, we have

$$f_1(\theta x + \beta y) \leq \theta f_1(x) + \beta f_1(y),$$
$$f_2(\theta x + \beta y) \leq \theta f_2(x) + \beta f_2(y).$$

Then, we examine $n = 3$, compute $f_3 = f_1 + f_2$, and obtain

$$\begin{aligned}
f_3(\theta x + \beta y) &= f_1(\theta x + \beta y) + f_2(\theta x + \beta y) \\
&\leq \theta f_1(x) + \theta f_2(x) + \beta f_1(y) + \beta f_2(y) \\
&\leq \theta(f_1(x) + f_2(x)) + \beta(f_1(y) + f_2(y)) \\
&\leq \theta f_3(x) + \beta f_3(y).
\end{aligned}$$

This means that $n = 3$ is established.

Assume this theorem holds for $n = k$, i.e.,

$$f_k(\theta x + \beta y) \leq \theta f_k(x) + \beta f_k(y),$$

and then verify the case for $n = k + 1$. We have

$$\begin{aligned}
f_{k+1}(\theta x + \beta y) &= f_k(\theta x + \beta y) + f_1(\theta x + \beta y) \\
&\leq \theta(f_k(x) + f_1(x)) + \beta(f_k(y) + f_1(y)) \\
&\leq \theta f_{k+1}(x) + \beta f_{k+1}(y).
\end{aligned}$$

This implies $n = k + 1$ is established.

Therefore, this theorem holds for any $\alpha > 0$ and $n \geq 1$. $\qquad\square$

*Proof of Lemma 3.* The proof of this lemma, concerning the convergence of the gradient descent method for strong convex functions, is classical and can be found in standard textbooks or papers on optimization theory. Therefore, we do not provide a specific reference. Alternatively, one may consult large language models, such as ChatGPT, DeepMind, or DeepSeek, to obtain the detailed proof. $\qquad\square$

*Proof of Lemma 4.* This is a classical concentration inequality, and the proof is omitted here. For more details, please refer to (McDiarmid et al., 1989). However, we need to note the following: originally, for McDiarmid's inequality and the setting in study (Bousquet & Elisseeff, 2002), the $z$ in $D$ and $D^v$ denotes a pair $(x, y)$. In our work, we consider $Z$ as $N$ pairs $(x, y)_{i=1}^n$. This setting does not introduce additional assumptions or requirements, as long as $R(h, D)$ can be regarded as a random variable. $\qquad\square$

*Proof of Lemma 5.* The proof can be found in (Bousquet & Elisseeff, 2002), on page 515, referring to the proof of Lemma 23. $\qquad\square$

A.4    EXPERIMENTS ON SYNTHETIC DATA

**Learning convergence analyses**: We exhibit the results in Figure 3, and these results are linked to the minimal $\beta$ for $\hat{h}$ learned by DLAERM. Based on the results, we observe that the learned models are generally close to the preset true model. This demonstrates that the stability bound is strongly connected to the generalization bound, indicating the effectiveness of our theoretical results for guiding learning under limited data conditions in domain generalization.

**Codes**: We provide the relevant codes for generating the synthetic data in Codes 1 and 2, and present the codes for DLAERM in Codes 3 and 4.

A.5    EXPERIMENTAL DETAILS

In this section, we mainly aim to validate the inversely proportional relationship between $\lambda$ and $EN$ through the order of the bound (12) ($\mathcal{O}(1/\sqrt{\lambda EN})$) and to assess the effectiveness of the regularization item in DLAERM. Here, we assume that for real-world datasets the $\beta^*$ to be a small constant. Specifically, (1) We can compare the $L_1$ norm of the weights for the trained models by

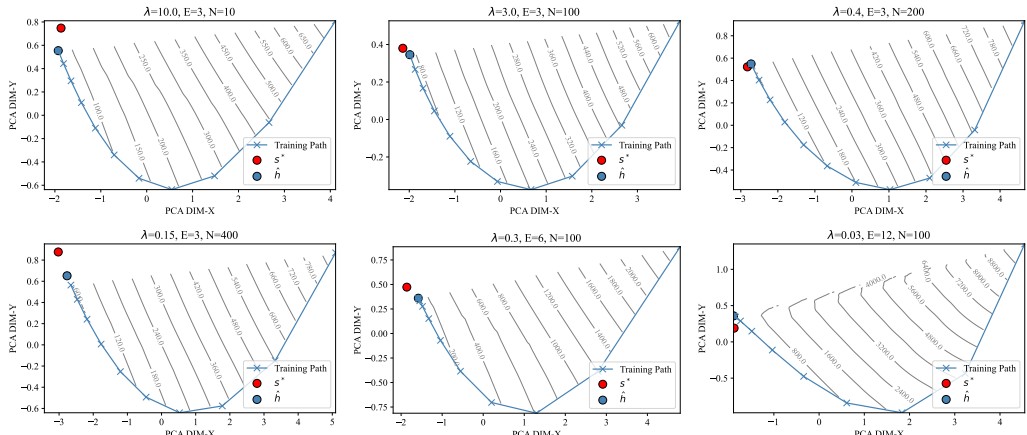

Figure 3: The learning convergence analyses of DLAERM with different $\lambda$, $E$, and $N$.

DLAERM (10) using datasets of different sizes. If the $L_1$ norm of a model is small when trained on a dataset with a small amount of data, i.e., $EN$ is small, we say that the corresponding $\lambda$ is large, demonstrating the inverse proportionality; and vice versa. (2) We can compare the classification results of models trained by ERM and DLAERM using the same dataset, and then observe the changes in evaluation metrics. If we observe an increase in accuracy in any domain after using the regularization term $\sum \|h\|_2^2$, we say that DLAERM performs better in domain generalization.

**Datasets**: Two image classification datasets were introduced to validate our learning methods: the PACS (Li et al., 2017) dataset and the OfficeHome (Venkateswara et al., 2017) dataset. The PACS dataset consists of four image domains: art painting, cartoon, photo, and sketch, while the OfficeHome dataset also includes four: art, clipart, product, and real-world. Namely, the PACS dataset consists of four image domains: $e_{art}$, $e_{cartoon}$, $e_{photo}$, and $e_{sketch}$, while the OfficeHome dataset also includes four: $e_{art}$, $e_{clipart}$, $e_{product}$, and $e_{photo}$. Typically, the last domain refers to the unseen domain $e_{unseen}$ and is excluded from the training domains in most studies of domain generalization; we follow this setting as well. We divided these datasets into training and testing sets, with only the testing sets containing the unseen domain data. Note that we extracted only SEVEN classes from the original OfficeHome dataset (which has sixty-five classes) to align with the number of classes in the PACS dataset. These classes in the OfficeHome datasets are alarm clock, bike, bottle, candles, chair, toys, and TV. This means that the classification problem in our experiments is a seven-class problem. Moreover, we did NOT use any data augmentation techniques. The PACS dataset is original, and we did not change it. The precise number of data samples in the two training datasets is as follows: for the PACS dataset, 1637 in art, 1873 in cartoon, and 1333 in photo; for the OfficeHome dataset, 522 in art, 366 in clipart, and 459 in product. Then, we have $N_{pacs} > N_{officehome}$ in each domain, which indicates that we need larger $\lambda$ values for learning methods using the OfficeHome training data in order to obtain considerable results.

**Classifier**: We incorporated only the well-known Resnet-50 architecture (He et al., 2016) and introduced the ImageNet weights for it. However, the CNN architecture is not included. This is because many studies have shown that the performance of Resnet-based models is better than that of CNN-based models in domain generalization.

**Metrics**: The primary evaluation metric for our experimental results is classification accuracy. A higher value indicates better performance for classifiers. Here, we use the $L_1$ norm of the weights to represent the complexity of trained models, as a lower $L_1$ norm indicates more zeros in the weights, i.e., simpler weights. We need to claim that a small $L_1$ norm of the weights indicates that the corresponding $\lambda$ is large in learning objectives.

**Loss Function**: We also use the mean squared error (MSE) as the loss function.

**Learning algorithms**: We incorporate the baseline ERM, LASERM, VREx (Krueger et al., 2021), and ERMPP (Teterwak et al., 2023), as well as the invariance-based IRM (Arjovsky et al., 2019), P-IRM (Choraria et al., 2023), and IB-IRM (Ahuja et al., 2021), and the non-invariance-based

---

Algorithm 1: Learning algorithm using our regularization.

---

**Input**: $D := \{D_e | e \in \mathcal{E}_{tr}\}$.
**Parameter**: Epoch, BatchSize: $b$.
**Output**: $h$.

1: **while** Epoch **do**
2:   **while** $N := \max\{n_e/b | e \in \mathcal{E}_{tr}\}$ **do**
3:     Let $\mathbf{d} \subset D_{tr}$.
4:     Let $L_{norm} = 0$.
5:     Let $R = 0$.
6:     **while** $\forall \mathbf{d}_e \in \mathbf{d}$ **do**
7:       Compute $\hat{Y}_b^e = h(X_b^e)$.
8:       Solve $R^e(h)$ using $\hat{Y}_b^e$ and $Y_b^e$.
9:       Solve $\sum_{i \in I} \lambda_i c_i(h)$ if needed.
10:       $L_{norm} \leftarrow L_{norm} + \|h\|_2^2$.
11:       $R \leftarrow R + R^e(h)$.
12:     **end while**
13:     Optimize $h$ using $R + \sum_{i \in I} \lambda_i c_i(h) + \lambda_{dla} L_{norm}$.
14:   **end while**
15: **end while**

1. $\mathbf{d} := \{\mathbf{d}_e | e \in \mathcal{E}_{tr}\}$, where $\mathbf{d}_e := \{(x_i^e, y_i^e)\}_{i=1}^b = (X_b^e, Y_b^e)$, $b \ll n_e$, i.e., $\mathbf{d}_e \subset D_e, \forall e \in \mathcal{E}_{tr}$, randomly sampled. Note that $Y_b^e = Y_b^{e'}, \forall e, e' \in \mathcal{E}_{tr}$ in $\mathbf{d}$.

2. $\sum_{i \in I} \lambda_i c_i(h)$ may vary based on different algorithms. If additional settings are required, please refer to the specific cited algorithms.

3. Here, $\mathcal{E}_{tr}$ enotes the training domain index set, i.e., $\mathcal{E}_{tr} = \{1, 2, \ldots\}$. For example, if the dataset $D$ contains 4 training domains, then $D := \{D_1, D_2, D_3, D_4\}$.

---

FISH (Shi et al., 2022), EQRM (Eastwood et al., 2022), and RDM (Nguyen et al., 2024). The rationale behind this choice is that these algorithms are open-source, widely recognized, and represent the state-of-the-art (SOTA). For more reliable results, we also incorporate $\sum \|h\|_2^2$ into VREX, ERMPP, FISH, EQRM, and RDM, and then add the prefix *DLA* to them, resulting in names such as DLAFISH. Details of these regularization-based methods, as well as LASERM, are provided in Appendix A.5. Note that we do not add regularization to the IRM-based methods due to conflicts between feature selection and invariance, which can be inferred from the complexity of their Hessian matrices.

**Hyperparameters for learning algorithms**: Note that all hyperparameters of VREx, ERMPP, IRM, P-IRM, IB-IRM, FISH, EQRM, and RDM are sourced from the DomainBed website[1].

**LASERM**: We denote this learning objective by $\min_h R(h) + \lambda_{las} \|h\|_1$.

**Algorithms** We provide a general learning algorithm, as shown in Algorithm 1. Note that in this algorithm, we use the following learning objective to replace Formula (10), which is:

$$\min_h L_{dla+}(\lambda_{dla}, \lambda, h) = \min_h R(h) + \lambda \sum_{i \in [E]} \|h\|_2^2 + \sum_{i \in I} \lambda_i c_i(h).$$

Here, $\sum_{i \in I} \lambda_i c_i(h)$ denotes the constraint from non-invariance-based learning methods, such as ERMPP, FISH, EQRM, etc. When $\sum_{i \in I} \lambda_i = 0$, The above formula reduces to DLAERM. The reason for introducing the constraint into other non-invariance-based learning methods is that we believe our regularization is a general constraint that can be regarded as an important norm for DG in real-world applications. Note that we did not incorporate our constraint into IRM-based methods due to conflicts between feature selection and invariance, which can be inferred from the complexity of their Hessian matrices.

**Experimental Conditions**: The entire set of experiments was performed on a single Nvidia RTX 4090 24 GB GPU, utilizing the Tensorflow-gpu 2.6.0 deep learning library.

---

[1]https://github.com/facebookresearch/DomainBed

Table 1: Classification accuracy (%) (mean ± std) on the two testing datasets achieved by Resnet50 trained using different learning algorithms. $e_{unseen}$ and $\mu$ denote the results for the unseen domain and the mean values across all domain results, respectively. The $\uparrow$ indicates improvement in evaluations compared to methods without the prefix. The Roman numerals denote three types of learning methods: IRM-based, ERM-based, and regularization-based methods.

| | Algorithms | PACS | | OfficeHome | |
|---|---|---|---|---|---|
| | | $e_{unseen}$ | $\mu$ | $e_{unseen}$ | $\mu$ |
| I | IRM | 40.4±0.5 | 51.5±1.3 | 55.6±3.4 | 59.2±2.9 |
| | PIRM | 40.4±1.2 | 51.7±1.3 | 54.8±2.8 | 60.0±3.1 |
| | IBIRM | 38.2±3.5 | 52.2±1.8 | 58.8±1.8 | 59.7±2.5 |
| II | ERM | 39.7±1.5 | 52.5±1.3 | 56.3±2.3 | 60.3±2.8 |
| | ERMPP | 38.9±2.0 | 52.4±1.5 | 55.5±2.3 | 59.4±3.0 |
| | FISH | 37.2±0.9 | 51.7±0.9 | 56.0±2.0 | 61.1±1.6 |
| | RDM | 36.7±3.1 | 48.7±2.2 | 58.0±2.7 | 61.3±2.3 |
| | VREX | 35.4±2.2 | 50.3±1.7 | 55.1±0.4 | 59.7±2.2 |
| | EQRM | 39.2±0.8 | 51.4±1.3 | 55.3±1.9 | 61.0±2.4 |
| III | LASERM $\uparrow$ | 40.0±2.6 | 52.8±1.6 | 58.2±1.8 | 62.4±1.8 |
| | DLAERM $\uparrow$ | 42.4±2.3 | 53.6±1.7 | 63.4±1.8 | 64.7±2.2 |
| | DLAERMPP $\uparrow$ | 41.6±3.1 | 53.6±2.1 | 60.3±1.2 | 64.8±2.5 |
| | DLAFISH $\uparrow$ | 40.7±2.3 | 52.8±1.6 | 61.0±3.3 | 64.7±2.7 |
| | DLARDM $\uparrow$ | 44.7±1.9 | 54.5±1.6 | 62.2±0.5 | 62.7±1.8 |
| | DLAVREX $\uparrow$ | 39.1±2.2 | 52.6±1.5 | 59.3±1.6 | 62.1±1.7 |
| | DLAEQRM $\uparrow$ | 39.4±0.9 | 52.3±1.0 | 60.7±0.9 | 63.2±2.3 |

**Computing Regularization**: In the experiments, we compute $\|h\|_1$ and $\|h\|_2^2$ as follows: $\|h\|_1 = \sum_{w \in h} \|w\|_1$ and $\|h\|_2^2 := \sum_{w \in h} \|w\|_2^2$.

**Training settings**: Since our experiment includes many learning algorithms, we fixed several training settings for all learning methods, classifiers, and datasets. These settings include the Adam optimizer, a learning rate of $1e-4$, and a batch size of 32. For ResNet50 architecture, we set 300 epochs on the PACS dataset and 500 epochs on the OfficeHome dataset. The introduced ResNet50 loads the provided parameters, and we only free the last convolution block in ResNet50 for training optimization. This means we perform some fine-tuning on the ResNet50 classifier. Moreover, the experiment for each learning algorithm and dataset was trained at least 10 times to ensure robust and reliable results.

**Main results**: Our main classification results are presented in Table 1, and the model complexity measured in the $L_1$ norm is shown in Figure 4. Here, Table 1 presents a summary of results, where we report only the evaluation in the unseen domain $e_{unseen}$ and the mean values across all domains. In Figure 4, the $L_1$ values of all models trained by each learning method are displayed; as observed, we use a box plot for visualization. For readers who are unfamiliar with the box plot, observing the orange line is sufficient. Conclusions: (a) The performance of models trained using the learning methods incorporating our regularization is improved, particularly when the data sample is limited. This is supported by the results in Table 1, for example, the results for the unseen domain on the OfficeHome testing data. (b) The inversely proportional relationship between $\lambda$ and $EN$ is established, i.e., the larger $EN$, the lower $\lambda$, and consequently, the higher $L_1$ norm value, and vice versa. This is supported by the results in Figure 4; in fact, we have $\lambda_{offichome} = 0.001$ and $\lambda_{pacs} = 1e-7$. Moreover, we can also observe the difference between using regularization and not using it.

**Hypothesis Testing**: To prove the first conclusion in the previous main results, we employ hypothesis testing. First, the following null hypothesis $H_0$ is presented, and the alternative hypothesis $H_1$ is our conclusion.

$H_0$ : The classification evaluations achieved by different learning algorithms and datasets are not statistically significantly different when inter-domain sparsity is constrained during learning.

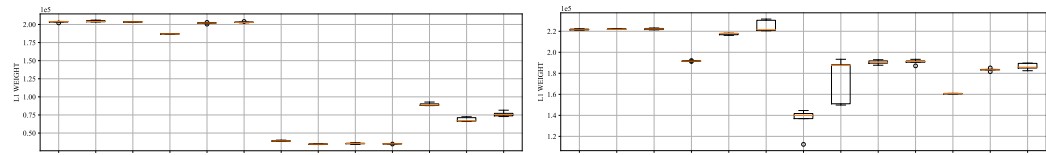

Figure 4: The $L_1$ norm of weights for trained models using different learning methods. The x-axis denotes the learning methods, while the y-axis represents the $L1$ value. The LEFT figure is the result using the OfficeHome data, while the RIGHT one is using the PACS data. We can clearly observe that the $L_1$ values for DLA-based methods on the LEFT are much smaller than the values for those methods on the RIGHT, which indicates that the $\lambda$ on the LEFT is larger than on the RIGHT.

Table 2: The hypothesis testing results for the data (excluding IRM-based and LASERM resluts) from Table 1. MWT denotes the Mann-Whitney U Test, KST stands for the K-S Two-sample Test, 'o' and 't' indicate one-sided and two-sided tests, and CD and AUC represent Cliff's Delta and Area Under the Curve, respectively. 's' and 'p' denote the statistic and p-value, respectively.

| $\alpha = 0.05$ | $Group_w$ | | $Group_0$ | | $Group_b$ | |
|---|---|---|---|---|---|---|
| | s | p | s | p | s | p |
| MWT, o | 387.5 | 0.020 | 390.0 | 0.018 | 394.0 | 0.014 |
| MWT, t | 387.5 | 0.041 | 390.0 | 0.036 | 394.0 | 0.029 |
| KST, o | 0.416 | 0.014 | 0.458 | 0.005 | 0.500 | 0.002 |
| KST, t | 0.416 | 0.029 | 0.458 | 0.011 | 0.500 | 0.004 |
| CD | 0.345 | / | 0.354 | / | 0.368 | / |
| AUC | 0.672 | / | 0.677 | / | 0.684 | / |

Second, we build the testing data. The evaluation results excluding IRM-based methods and LASERM in Table 1 can be divided into two arrays without considering specific datasets: $g0$ and $g1$. The $g0$ array contains all results from ERM to EQRM, while $g1$ includes all results from DLAERM to DLAEQRM. Based on the mean values and standard deviations, we can build three types of data groups: the worst evaluation group $Group_w$, the mean evaluation group $Group_0$, and the best evaluation group $Group_b$. For example, $Group_w : \{g0_w, g1_w\}$, where $g0_w = \{39.7 - 1.5, 52.5 - 1.3, \dots, 55.3 - 1.9, 61.0 - 2.4\}$ and $g1_w = \{42.4 - 2.3, 53.6 - 1.7, \dots, 60.7 - 0.9, 63.2 - 2.3\}$, $Group_0 : \{g0_0, g1_0\}$, where $g0_0 = \{39.7, 52.5, \dots, 55.3, 61.0\}$ and $g1_0 = \{42.4, 53.6, \dots, 60.7, 63.2\}$, and $Group_b := \{g0_b, g1_b\}$, where $g0_b = \{39.7 + 1.5, 52.5 + 1.3, \dots, 55.3 + 1.9, 61.0 + 2.4\}$ and $g1_b = \{42.4 + 2.3, 53.6 + 1.7, \dots, 60.7 + 0.9, 63.2 + 2.3\}$. Consequently, we have three new null hypotheses derived from $H_0$, as follows:

1. $H_{0,group_w}$: $g1_w$ is not significantly different from $g0_w$.

2. $H_{0,group_0}$: $g1_0$ is not significantly different from $g0_0$.

3. $H_{0,group_b}$: $g1_b$ is not significantly different from $g0_b$.

Now, we reject $H_0$ if $H_{0,group_w}$, $H_{0,group_0}$, and $H_{0,group_b}$ are all rejected. Third, based on $Group_w$, $Group_0$, and $Group_b$, we incorporate the Mann-Whitney U Test and the K-S Two-sample Test, as the data do not follow Gaussian distributions. For convenience, precision, and reproducibility, we utilize the python library "scipt.stats" to compute statistics and p-values, with all results shown in Table 2. Moreover, we compute Cliff's Delta and the Area Under the Curve to enhance the test results. Finally, based on the significance level $\alpha = 0.05$, we can reject both $H_{0,group_w}$, $H_{0,group_0}$, and $H_{0,group_b}$, as well as $H_0$. This means our conclusion is statistically established.

**Details of classification results**: We provide Tables 3 and 4. As observed, the results obtained by our learning methods on both the PACS and OfficeHome datasets are improved, regardless of known or unseen domains. These results provide an interesting observation: the performance of models with more zero weights is better than that of models with fewer zero weights under the same architecture. This suggests that, under real-world data conditions, unless the dataset is very large, most of the weights in the network architecture are unnecessary and can be replaced by zero. This further implies that many weights in larger models may also be redundant.

Table 3: Classification accuracy (%) on the PACS testing dataset.

| | $e_{art}$ | $e_{cartoon}$ | $e_{photo}$ | $e_{sketch}$ | $\mu$ |
|---|---|---|---|---|---|
| IRM | 41.3±1.3 | 61.6±1.6 | 62.7±1.9 | 40.4±0.5 | 51.5±1.3 |
| PIRM | 42.3±2.0 | 61.6±0.9 | 62.4±1.3 | 40.4±1.2 | 51.7±1.3 |
| IBIRM | 43.9±1.5 | 63.4±0.9 | 63.1±1.2 | 38.2±3.5 | 52.2±1.8 |
| ERM | 41.4±1.8 | 62.8±0.8 | 66.0±1.2 | 39.7±1.5 | 52.5±1.3 |
| ERMPP | 42.5±1.3 | 63.1±1.1 | 65.2±1.4 | 38.9±2.0 | 52.4±1.5 |
| FISH | 41.8±1.4 | 62.5±1.0 | 65.3±0.3 | 37.2±0.9 | 51.7±0.9 |
| RDM | 39.6±1.9 | 57.5±1.7 | 60.9±2.2 | 36.7±3.1 | 48.7±2.2 |
| VREX | 39.5±2.0 | 61.5±1.5 | 64.9±0.9 | 35.4±2.2 | 50.3±1.7 |
| EQRM | 40.6±2.4 | 62.2±0.6 | 63.7±1.3 | 39.2±0.8 | 51.4±1.3 |
| LASERM↑ | 42.9±2.0 | 63.5±1.2 | 64.9±0.6 | 40.0±2.6 | 52.8±1.6 |
| DLAERM↑ | 43.0±1.4 | 63.9±1.5 | 64.9±1.3 | 42.4±2.3 | 53.6±1.7 |
| DLAERMPP↑ | 42.7±1.7 | 65.9±2.1 | 64.0±1.3 | 41.6±3.1 | 53.6±2.1 |
| DLAFISH↑ | 41.5±0.9 | 64.1±2.4 | 64.7±0.8 | 40.7±2.3 | 52.8±1.6 |
| DLARDM↑ | 45.0±2.7 | 63.1±1.1 | 65.5±0.9 | 44.7±1.9 | 54.5±1.6 |
| DLAVREX↑ | 43.2±1.9 | 62.5±1.3 | 65.5±0.7 | 39.1±2.2 | 52.6±1.5 |
| DLAEQRM↑ | 42.3±0.8 | 63.1±1.0 | 64.5±1.2 | 39.4±0.9 | 52.3±1.0 |

Table 4: Classification accuracy (%) on the OfficeHome testing dataset.

| | $e_{art}$ | $e_{clipart}$ | $e_{product}$ | $e_{photo}$ | $\mu$ |
|---|---|---|---|---|---|
| IRM | 49.3±3.0 | 63.8±2.9 | 68.2±2.4 | 55.6±3.4 | 59.2±2.9 |
| PIRM | 51.1±2.5 | 65.5±3.9 | 68.8±3.2 | 54.8±2.8 | 60.0±3.1 |
| IBIRM | 49.3±5.0 | 65.3±1.1 | 65.3±1.9 | 58.8±1.8 | 59.7±2.5 |
| ERM | 48.7±3.0 | 63.6±1.7 | 72.6±3.9 | 56.3±2.3 | 60.3±2.8 |
| ERMPP | 47.8±5.8 | 62.7±1.8 | 71.6±2.2 | 55.5±2.3 | 59.4±3.0 |
| FISH | 49.1±2.2 | 63.6±2.2 | 75.8±0.0 | 56.0±2.0 | 61.1±1.6 |
| RDM | 51.5±3.7 | 64.4±1.9 | 71.2±1.1 | 58.0±2.7 | 61.3±2.3 |
| VREX | 50.0±2.6 | 63.6±2.6 | 70.1±3.2 | 55.1±0.4 | 59.7±2.2 |
| EQRM | 53.0±2.2 | 62.9±2.7 | 72.8±2.7 | 55.3±1.9 | 61.0±2.4 |
| LASERM↑ | 53.5±2.8 | 62.9±1.7 | 75.2±0.8 | 58.2±1.8 | 62.4±1.8 |
| DLAERM↑ | 54.8±3.7 | 62.7±1.2 | 77.9±2.1 | 63.4±1.8 | 64.7±2.2 |
| DLAERMPP↑ | 56.1±3.3 | 64.1±2.7 | 78.7±2.6 | 60.3±1.2 | 64.8±2.5 |
| DLAFISH↑ | 56.1±2.8 | 65.5±1.5 | 76.4±3.3 | 61.0±3.3 | 64.7±2.7 |
| DLARDM↑ | 53.0±2.5 | 62.7±2.2 | 72.8±1.8 | 62.2±0.5 | 62.7±1.8 |
| DLAVREX↑ | 53.4±2.2 | 63.3±1.5 | 72.2±1.6 | 59.3±1.6 | 62.1±1.7 |
| DLAEQRM↑ | 56.1±3.2 | 62.4±1.8 | 73.5±3.1 | 60.7±0.9 | 63.2±2.3 |

**Ablation Study & Hyperparameters experiments for $\lambda_{las}$ and $\lambda_{dla}$:** We can consider the comparison between ERM, LASERM, and DLAERM as an ablation study, and the results are shown in Figures 5 and 6. Figure 5 displays the results on the PACS testing data obtained by the baseline (ERM), LASERM (T1), and DLAERM (T2). The left figure shows the results for the unseen domain, while the right one presents the average results across all domains. Figure 6 displays the results on the OfficeHome testing data, using the same settings as Figure 5.

Note that this experiment also exhibits the choice of hyperparameters ($\lambda_{las}$ and $\lambda_{dla}$). We display the detail hyperparameter experimental results for $\lambda_{las}$ in Tables 5 and 6, and the results for $\lambda_{dla}$ in Tables 7 and 8. For different datasets, we selected different values for $\lambda_{las}$ and $\lambda_{dla}$. As observed, although more optimal results may exist, we selected the values based on the performance in the unseen domain.

Based on all results, we conclude that DLAERM performs better, regardless of the domains and datasets. Moreover, we find that $\lambda_{dla}$ is larger when the number of data samples is relatively low, and vice versa. For example, $1e-3$ for the OfficeHome data, and $1e-7$ for the PACS data.

**The $L_1$ norm for LASERM and DLAERM:** Based on the different hyperparameters for LASERM and DLAERM, we present the sparsity of all trained models in Figures 7 and 8. In each figure, the

Table 5: Hyperparameter $\lambda_{las}$ experiments on the PACS testing dataset achieved by Resnet50. The * indicates the final selection. The red denotes the best results.

| $\lambda_{las}$ | $e_{art}$ | $e_{cartoon}$ | $e_{photo}$ | $e_{sketch}$ | $\mu$ |
|---|---|---|---|---|---|
| ERM | 41.4±1.8 | 62.8±0.8 | 66.0±1.2 | 39.7±1.5 | 52.5±1.3 |
| 0.001 | 19.9±1.7 | 16.8±0.3 | 17.1±7.1 | 13.4±7.6 | 16.8±4.2 |
| 0.0005 | 25.5±4.1 | 32.6±3.3 | 33.4±8.6 | 22.5±8.2 | 28.5±6.1 |
| 0.0001 | 38.9±2.0 | 63.3±1.7 | 67.7±2.2 | 37.4±3.1 | 51.8±2.2 |
| 5e-05 | 44.6±3.4 | 66.5±1.7 | 68.1±1.4 | 35.8±2.2 | 53.8±2.2 |
| 1e-05 | 43.4±2.1 | 66.3±1.8 | 66.0±1.4 | 39.6±2.2 | 53.8±1.9 |
| 5e-06 | 43.6±1.7 | 66.2±1.3 | 66.2±1.7 | 39.3±2.4 | 53.8±1.8 |
| *1e-06 | 42.9±2.0 | 63.5±1.2 | 64.9±0.6 | 40.0±2.6 | 52.8±1.6 |

Table 6: Hyperparameter $\lambda_{las}$ experiments on the OfficeHome testing dataset achieved by Resnet50. The * indicates the final selection.

| $\lambda_{las}$ | $e_{art}$ | $e_{clipart}$ | $e_{product}$ | $e_{photo}$ | $\mu$ |
|---|---|---|---|---|---|
| ERM | 48.7±3.0 | 63.6±1.7 | 72.6±3.9 | 56.3±2.3 | 60.3±2.8 |
| 0.01 | 14.6±3.3 | 27.1±6.3 | 24.4±7.6 | 21.0±2.5 | 21.8±4.9 |
| 0.005 | 14.1±3.6 | 31.2±4.2 | 26.7±6.2 | 23.4±3.5 | 23.9±4.4 |
| 0.001 | 40.9±4.5 | 58.6±2.2 | 64.2±2.9 | 52.1±1.5 | 54.0±2.8 |
| 0.0005 | 48.0±1.4 | 59.4±3.3 | 70.9±2.5 | 56.6±2.0 | 58.8±2.3 |
| *0.0001 | 53.5±2.8 | 62.9±1.7 | 75.2±0.8 | 58.2±1.8 | 62.4±1.8 |
| 5e-05 | 52.3±4.9 | 64.2±1.3 | 77.1±3.4 | 57.7±2.6 | 62.8±3.1 |
| 1e-05 | 49.3±2.2 | 63.9±3.1 | 73.7±3.3 | 55.5±2.3 | 60.6±2.8 |
| 5e-06 | 48.9±2.2 | 64.5±2.8 | 72.8±2.4 | 57.3±3.3 | 60.9±2.7 |
| 1e-06 | 51.5±3.6 | 60.2±5.1 | 68.0±9.2 | 52.7±5.4 | 58.1±5.8 |

upper part shows the results of LASERM, while the lower part shows the results of DLAERM. Comparing these results, we find that the $L_1$ norm of DLAERM is higher than that of LASERM under the same hyperparameters, which suggests that the inter-domain sparsity is a relaxed version of the global one.

**Explanations for Table 1**: The domain-aware specific evaluation results are exhibited in Tables 3 and 4, and we only exhibit the results of the unseen domain and the mean values of all domains. Note that we do not intend to declare which learning method is best. Here, we conclude that the performance of IRM-based methods shows no significant difference compared to ERM-based methods on real-world datasets. Thus, we argue that there is still a long way to go in developing a learning method that can truly discover invariant features under complex real-world data conditions. For practical applications, we still recommend using ERM-based methods combined with our proposed constraint.

**Explanations for Table 2**: First, we explain the one-sided testing for MWT and KST. Based on the hypotheses $H_{0,group_w}$, $H_{0,group_0}$, and $H_{0,group_b}$, the one-sided testing specifically refers to the $g1 > g0$ test. The explanations of Cliff's Delta $s_{cd} \in [-1, 1]$: if $s_{cd} = 1$, $g1 \gg g0$; if $s_{cd} = -1$, $g1 \ll g0$; and if $s_{cd} = 0$, $g1 \approx g0$. The values of $s_{cd}$ from this table belong to the medium range $[0.33, 0.474]$, indicating $g1 > g0$ for both data groups. The explanations of AUC $s_{auc} \in [0, 1]$: if $s_{auc} = 0.5$, $g1 \approx g0$; if $s_{auc} > 0.5$, $g1 > g0$; if $s_{auc} < 0.5$, $g1 < g0$; if $s_{auc} = 0.0$, $g1 \ll g0$; and if $s_{auc} = 1.0$, $g1 \gg g0$. The values of $s_{auc}$ from this table are greater than 0.5, indicating $g1 > g0$ for both data groups.

Table 7: Hyperparameter $\lambda_{dla}$ experiments on the PACS testing dataset achieved by Resnet50. The * indicates the final selection.

| $\lambda_{dla}$ | $e_{art}$ | $e_{cartoon}$ | $e_{photo}$ | $e_{sketch}$ | $\mu$ |
|---|---|---|---|---|---|
| ERM | 41.4±1.8 | 62.8±0.8 | 66.0±1.2 | 39.7±1.5 | 52.5±1.3 |
| 0.001 | 36.8±1.6 | 56.4±1.5 | 65.2±2.1 | 36.2±3.4 | 48.6±2.1 |
| 0.0005 | 41.9±0.7 | 65.6±2.3 | 67.1±2.4 | 39.2±2.4 | 53.5±2.0 |
| 0.0001 | 43.2±2.2 | 66.0±2.5 | 65.5±1.6 | 39.3±1.4 | 53.5±1.9 |
| 5e-05 | 42.6±1.0 | 65.5±0.6 | 65.5±0.4 | 41.3±2.9 | 53.7±1.3 |
| 1e-05 | 42.7±0.9 | 64.7±1.8 | 65.0±1.1 | 41.5±2.1 | 53.5±1.5 |
| 5e-06 | 40.0±2.7 | 62.1±3.6 | 64.5±2.6 | 40.2±2.5 | 51.7±2.9 |
| 1e-06 | 42.6±0.5 | 65.5±1.0 | 64.9±1.0 | 40.1±1.5 | 53.3±1.0 |
| 5e-07 | 44.1±1.8 | 64.6±1.0 | 64.6±1.6 | 41.2±1.3 | 53.6±1.4 |
| *1e-07 | 43.0±1.4 | 63.9±1.5 | 64.9±1.3 | 42.4±2.3 | 53.6±1.7 |

Table 8: Hyperparameter $\lambda_{dla}$ experiments on the OfficeHome testing dataset achieved by Resnet50. The * indicates the final selection.

| $\lambda_{dla}$ | $e_{art}$ | $e_{clipart}$ | $e_{product}$ | $e_{photo}$ | $\mu$ |
|---|---|---|---|---|---|
| ERM | 48.7±3.0 | 63.6±1.7 | 72.6±3.9 | 56.3±2.3 | 60.3±2.8 |
| 0.005 | 30.0±1.5 | 46.1±3.7 | 53.9±4.6 | 47.6±5.1 | 44.4±3.7 |
| *0.001 | 54.8±3.7 | 62.7±1.2 | 77.9±2.1 | 63.4±1.8 | 64.7±2.2 |
| 0.0005 | 52.7±0.9 | 63.0±1.8 | 76.6±1.5 | 63.3±0.9 | 63.9±1.3 |
| 0.0001 | 49.1±2.1 | 64.5±2.3 | 73.5±2.7 | 57.8±2.6 | 61.2±2.4 |
| 5e-05 | 50.2±4.8 | 64.2±1.2 | 73.1±3.7 | 57.6±2.5 | 61.3±3.0 |
| 1e-05 | 48.7±2.5 | 64.1±3.5 | 74.3±3.8 | 56.3±3.4 | 60.9±3.3 |
| 5e-06 | 49.6±2.9 | 64.2±1.5 | 74.3±2.1 | 55.3±1.2 | 60.9±1.9 |
| 1e-06 | 50.7±1.9 | 64.7±2.7 | 72.6±4.3 | 55.5±2.7 | 60.9±2.9 |

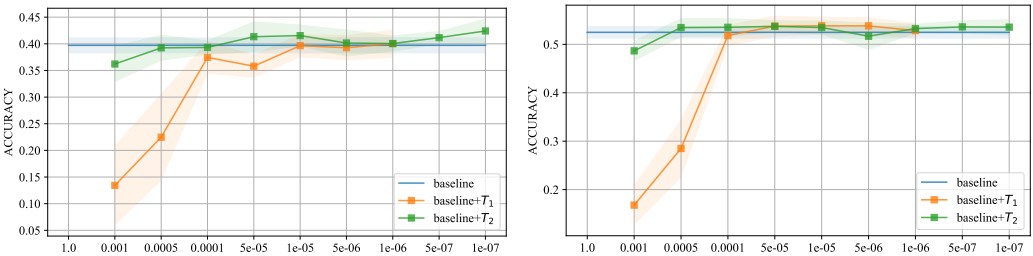

Figure 5: Ablation Study and Hyperparameter Comparisons conducted on the PACS testing data. For the ablation analysis, we report baseline results from ERM, T1 results from LASERM, and T2 results from DLAERM. For the hyperparameter comparison, the x-axis denotes different values, while the y-axis represents the evaluation metrics. The left figure shows the results for the unseen domain, while the right one presents the average results across all domains.

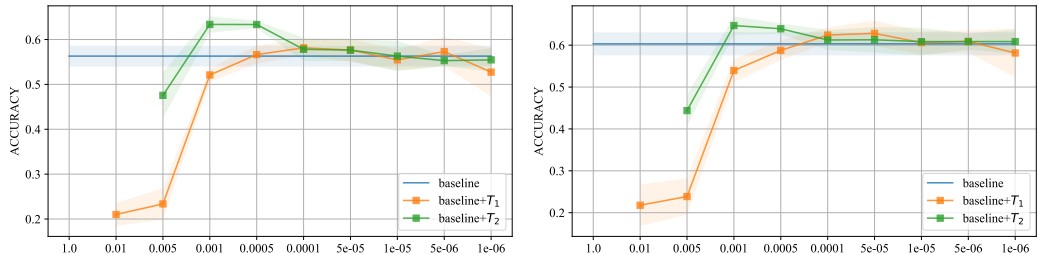

Figure 6: Ablation Study and Hyperparameter Comparisons conducted on the OfficeHome testing data. This figure uses the same settings as Figure 5.

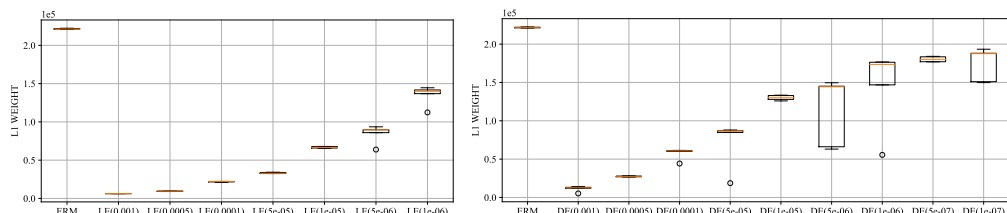

Figure 7: The $L_1$ norm of weights for the trained models using LASERM and DLAERM on the PACS testing data. The x-axis denotes different hyperparameters, while the y-axis represents the $L_1$ value. The left figure shows the results from LASERM, while the right one presents the results from DLAERM.

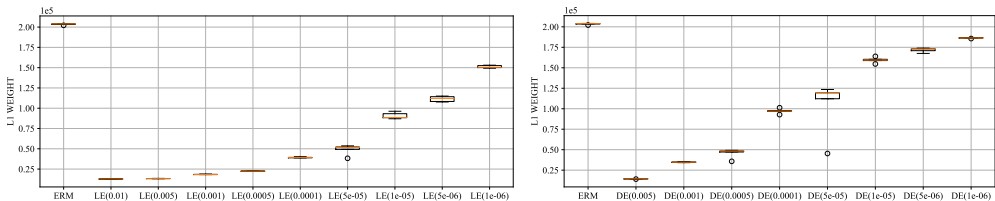

Figure 8: The $L_1$ norm of weights for the trained models using LASERM and DLAERM on the OfficeHome testing data. The x-axis denotes different hyperparameters, while the y-axis represents the $L_1$ value. The left figure shows the results from LASERM, while the right one presents the results from DLAERM.

---

Listing 1: Codes for generating synthetic data. [Part 1]

```
1   import numpy as np
2   import os
3   import pandas as pd
4
5
6   def DataSamplesClassification(gamma, n=100, d=2, env=1.0):
7       zc = np.random.randn(n, d)
8
9       noise = np.random.randn(n, d)
10      dot_product = np.sum(zc * noise)
11      noise_orthogonal = noise - dot_product * zc / np.sum(zc * zc)
12
13      ye = gamma * zc + noise_orthogonal * 1.0  # setting the weight of
            noise
14
15      ze = ye + np.random.randn(n, d) * env
16
17      return np.hstack([zc, ze, np.sum(ye, axis=1, keepdims=True)])
18
19
20  def create_load_gama(d, path, rewrite_gamma=False):
21      if rewrite_gamma:
22          print('Rewrite gama ...')
23          gamma = np.random.randn(d)
24          df_w = pd.DataFrame(np.hstack([gamma, np.zeros_like(gamma)]))
25          name = os.path.join(path, 'true_gama.csv')
26          df_w.to_csv(name, index=False)
27          print(name + ' has been built!')
28      else:
29          gamma = pd.read_csv(os.path.join(path, 'true_gama.csv')).
                values
30          gamma = np.squeeze(gamma[0: d])
31      return gamma
```

Listing 2: Codes for generating synthetic data. [Part 2]

```python
def create_data(num, d, path, rewrite_data=False, rewrite_gamma=False):
    # generate a random or load the old one

    if not os.path.exists(path):
        os.makedirs(path)

    gamma = create_load_gama(d, path, rewrite_gamma)
    Data_list = []

    if rewrite_data:

        print('Rewrite data ...')
        E = 3    # change here for domain number !!!!!!!!!!!!
        envs = np.arange(0, E+1) + 1.0    # for D^v

        for i, e in enumerate(envs):
            data = DataSamplesClassification(gamma, n=num, d=d, env=e)
            Data_list.append(np.mean(data, axis=0))

            df = pd.DataFrame(data)
            name = os.path.join(path, f'env{i + 1}.csv')
            df.to_csv(name, index=False)
            print(name + ' has been built!')

    long = np.shape(gamma)[0]
    expand_gamma = np.zeros(long*2)
    expand_gamma[0:long] = gamma

    return d * 2, expand_gamma  # this is the data dimension d*2

if __name__ == "__main__":
    data_root = '/preprocess_data'
    num, d = 200, 20  # change here for data number !!!!!!!!!!!!
    a = create_data(num, d, data_root, rewrite_data=True,
        rewrite_gamma=False)
```

Listing 3: Codes for Algorithms. [Part 1]

```python
import tensorflow as tf

class ERM(Algorithm):
    def __init__(self, dimension, num_domains, hparams):
        super(ERM, self).__init__(dimension, num_domains, hparams)

        self.w = tf.Variable(tf.random.normal(shape=(1, dimension)),
                             trainable=True, dtype=tf.float32,
                             name='W')     # (d, 1)

        self.loss = losses.MSE  # This MSE you can define by yourslef.
        self.optimizer = tf.keras.optimizers.Adam(learning_rate=self.
            hparams['lr'],
                                                  decay=self.hparams['
                                                      learning_decay'
                                                      ])

    @tf.function
    def update(self, sub_envs, unlabeled=None):
        loss = 0.
        with tf.GradientTape() as tape:
            for env in sub_envs:
                e_x, e_y = env
                predictions = tf.reduce_sum(self.w * e_x, axis=1)

                loss_e = self.loss(e_y, predictions)
                loss += tf.reduce_mean(loss_e)

        gradients = tape.gradient(loss, [self.w])
        self.optimizer.apply_gradients(zip(gradients, [self.w]))
        return tf.reduce_mean(loss)

    @tf.function
    def predict(self, x):
        predictions = tf.reduce_sum(self.w * x, axis=1)
        return predictions

    @tf.function
    def validate(self, sub_envs):
        loss = 0.
        for env in sub_envs:
            e_x, e_y = env
            predictions = self.predict(e_x)
            loss_e = self.loss(e_y, predictions)
            loss += tf.reduce_mean(loss_e)
        return tf.reduce_mean(loss)

    def stability_validate(self, gamma, sub_envs):
        loss = 0.
        for env in sub_envs:
            e_x, e_y = env
            predictions = tf.reduce_sum(gamma * e_x, axis=1)
            loss_e = self.loss(e_y, predictions)
            loss += tf.reduce_mean(loss_e)
        return tf.reduce_mean(loss)

    def squared_l2_norm(self, x):
        return tf.reduce_sum(tf.square(x))
```

Listing 4: Codes for Algorithms. [Part 2]

```python
class DLA(ERM):
    def __init__(self, dimension, num_domains, hparams):
        super(DLA, self).__init__(dimension, num_domains, hparams)

    @tf.function
    def update(self, sub_envs, unlabeled=None):
        loss = 0.
        sql2 = 0.
        penalty = self.hparams['dla_lambda']
        with tf.GradientTape() as tape:
            for env in sub_envs:
                e_x, e_y = env
                predictions = tf.reduce_sum(self.w * e_x, axis=1)

                loss_e = self.loss(e_y, predictions)
                loss += tf.reduce_mean(loss_e)
                sql2 += tf.add_n([self.squared_l2_norm(tf.cast(v, tf.
                    float32)) for v in [self.w]])

            loss += penalty*sql2

        gradients = tape.gradient(loss, [self.w])
        self.optimizer.apply_gradients(zip(gradients, [self.w]))
        return tf.reduce_mean(loss)
```

