# OpenReview forum: "Stability Bounds for Domain Generalization under Limited Data"
_ICLR.cc/2026/Conference — Submitted to ICLR 2026_

### Official Review · Reviewer_mfmp · 2025-10-18

**Soundness:** 2
**Presentation:** 2
**Contribution:** 3
**Rating:** 4
**Confidence:** 3

**Summary:**

This paper addresses limited-data domain generalization by deriving stability-based generalization bounds and proposing DLAERM, a domain-level L2-regularized method. The theory shows error decays exponentially with stability, and the algorithm ensures good generalization even with few domains, validated by theoretical guarantees and experiments.

**Strengths:**

- The paper derives new, exponential-decay generalization bounds for DG using stability theory. These bounds do not rely only on i.i.d. data.

- A new algorithm, DLAERM, is proposed. The method is proven to have a linear convergence rate, ensuring efficient optimization.

- Experiments on benchmarks (PACS, OfficeHome) confirm the theory.

**Weaknesses:**

- Most proofs appear valid, but Theorem 1 requires caution; the inequality in Eq.(3) may still depend on the number of domains $E$.

- The paper does not define a proper partial order on function spaces; for example, when writing $h \ge h'$, the meaning is not specified.

- Paper structure is complete and the main thread is clear, but notation and semantic roles (e.g., $R,R',R_{\exp}$) switch without clear clarification, increasing cognitive burden.

- Experimental interpretation is insufficient. Figure 1 shows trends but is not explicitly linked to theoretical quantities such as $\beta$.

- Missing discussion of classic and recent DG literatures, e.g.

[1] Gulrajani & Lopez-Paz (2021) In Search of Lost Domain Generalization, ICLR.

[2] Tong et al. (2023) Distribution Free Domain Generalization, ICML.

**Questions:**

1. Page 14, Lines 733. In the proof of Theorem 1, should the right hand side of the inequality be multiplied by an additional factor $E$? If so, this modification would propagate to the main results and may alter the final bounds.

2. The current proof of Lemma 1 is not mathematically rigorous with respect to the use of order relations and the treatment of subgradients.

3. (If possible) Demonstrating empirical performance on large scale datasets would further strengthen the credibility of the proposed method.

4. See Weaknesses

---

> ### Author Response · Authors · 2025-11-20
>
> Thank you for taking the time to review our work. Your comments are extremely valuable for improving it, and we greatly appreciate your careful scrutiny.
>
>
> For the proofs of Theorem 1 and Lemma 1, the related issues have been corrected in the revision. Specifically, we added the missing $E$ in the proof of Theorem 1, as shown below: $$ \sum\limits _{v \in [E]} {\mathop {\sup }\limits _{{{Z'} _v} \in {\cal Z}} \left| {R(h,D) - R(h,{D^v})} \right|}  \le E\beta .$$ Correspondingly, the upper bound in Theorem 1 has been revised to $R - {R^{\exp }} \le \sqrt {((E\beta )/2)\ln (1/\delta )}$, along with the related inequalities that follow. We modified the statement $ {h^ * } + t\Delta h \ne {h^ * }$ in the proof of Lemma 1, and we clarified that the subgradients $0$ and $g$ are elements of the hypothesis class rather than scalar values.
>
> As you noted, some results have been updated, although these changes do NOT affect our main argument. We hope you can review this revised version and provide us with further suggestions.
>
> Regarding the experimental suggestions, we conduct experiments on (1) synthetic data to validate the stability bounds $\beta$ and the learning convergence rate by varying $E$, $N$, and $\lambda$, and (2) real-world data for additional validation, including experiments with transformer-based backbones and with large datasets where $E$ and $N$ are manually controlled. We will update the results during the rebuttal phase to the best of our ability.
>
> The discussion of related DG literatures has been provided in Section 2.4. Specifically, we discussed the work by Tong et al., including their upper bound and its related limitations. Regarding the work by Gulrajani and Lopez-Paz, it was further discussed in the work by Wang et al. [1], which has already been addressed in our study. Therefore, we only briefly mention their work.
>
> Finally, we would like to thank you again, as your comments have greatly helped us improve our work, and we will acknowledge these contributions.
>
>
> [1] Wang, Y., Wu, Y. and Zhang, H. Lost Domain Generalization Is a Natural Consequence of Lack of Training Domains. In Proceedings of the AAAI Conference on Artificial Intelligence. 2024, March.

---

> > ### Comment · Reviewer_mfmp · 2025-11-21
> >
> > Thanks for the response. I will raise my socre to 6.

---

> > > ### Author Response · Authors · 2025-11-25
> > >
> > > Dear Reviewer,
> > >
> > > We have updated the experiments in the main paper, presenting results on synthetic data that are frequently used in domain generalization studies, for validating the stability bounds $\beta$ and analyzing learning convergence. Due to our limited experimental resources, the additional validations on real-world datasets cannot be provided during this rebuttal phase. Despite this, the flexibility of synthetic data in controlling $E$ and $N$ enables us to provide meaningful observations.
> > >
> > > The experimental results clearly exhibit: (1) the inverse proportional relationship between $\lambda$ and $EN$, (2) the optimal stability leading to good convergence, (3) the effectiveness of our method in addressing the less-sample problem in domain generalization, and (4) the tradeoff point for $\lambda$. All these results reinforce the significance of our contributions to domain generalization and address the limitations of our initial experiments.
> > >
> > >
> > > We hope you can review this revised version and provide us with further suggestions.
> > >
> > >
> > > Best,
> > >
> > > Authors.

---

### Official Review · Reviewer_oqPD · 2025-10-25

**Soundness:** 2
**Presentation:** 2
**Contribution:** 2
**Rating:** 2
**Confidence:** 3

**Summary:**

The paper studies domain generalization (DG) via algorithmic stability. It adapts the classical Bousquet–Elisseeff framework (with McDiarmid’s inequality) to a multi-domain setting by assuming independence (but not necessarily identical distribution) across domains, and then instantiates the bound for RKHS ERM with $\ell_2$-regularization. The main statement yields a stability parameter of the form
$\beta \le \tfrac{2\sigma^2\kappa^2}{\lambda E N}+\beta^\*$,
leading to a tail bound of McDiarmid type (“exponential decay”) and a convergence result under strong convexity and plain GD.

**Strengths:**

- The paper is well-structured; the stability-to-generalization pipeline is explicit and easy to follow.

- Instantiated stability for RKHS ERM. Provides a closed-form stability bound in terms of $(\lambda, E, N, \kappa, \sigma)$ that is interpretable and matches classical intuition (regularization improves stability).

- Simple empirical illustration. The toy experiments align with the qualitative prediction that stronger regularization helps in small-sample regimes.

**Weaknesses:**

- Limited novelty vs. classical stability. The core technical path (uniform stability $+$ McDiarmid tail) closely mirrors standard results; the “exponential-decay bound” is the usual concentration tail, not a new rate or DG-specific mechanism.

- Strong/atypical assumptions. The bespoke stability notion on $R(h,D)$ vs. $R(h,D_v)$, existence of $h^\*$ minimizing $|R-R'|$ with $\nabla R(h^\*)\ge0$, bounded kernel $\kappa$, and $\sigma$-admissibility collectively make the scope narrow; the role and size of $\beta^\*$ are left largely implicit.

- Theory–practice mismatch. Convergence is proved under strong convexity with plain GD, whereas experiments use deep models where assumptions do not hold; no bridge (e.g., surrogate convex surrogates or local strong convexity) is provided.

**Questions:**

Clarify the necessity and typical size of $\beta^\*$; provide conditions under which $\beta^\*\approx0$ or explicit upper bounds.

---

> ### Author Response · Authors · 2025-11-20
>
> Thank you for taking the time to review our work, and we greatly appreciate your scrutiny.
>
> Regarding novelty, we consider it essentially linked to the approach for addressing the main problem rather than the tools used. Our novelty essentially lies in providing a theoretical guarantee for SRM in DG when facing the less-sample problem via using the tools of the stability concept and McDiarmid’s inequality. The differences between our results and classical stability had already been discussed in the manuscript, including the definitions of error stability, the data settings, and the parameter $\beta$.
>
> Under our assumptions, we consider them to be mild. For example, assuming $ h^*  = \arg {\min _{h' \in {\cal H}}}\left| {R(h') - R'(h')} \right|$ is milder that assuming $ a = \arg {\min _{h' \in {\cal H}}}R(h')$ and $ b = \arg {\min _{h' \in {\cal H}}}R'(h')$ in DG, since we cannot guarantee that $a$ or $b$ leads to minimal stability $\beta^*$. Other notations are typically standard requirements.
>
> We do not specify the type of hypothesis classes in our theorems; therefore, the results remain valid for deep models, whose parameters can be updated using gradient-based methods.
>
> Finally, the condition for $\beta ^* \to 0$ is the existence of $h^*$. This constant serves as a tradeoff in dataset selection, analogous to the role of $\lambda$ in Ben-David’s bound [1]:
> $$ {{\epsilon} _T}(h) - {{\epsilon} _S}(h) \le \frac{1}{2}{{\hat d} _{{\cal H}\Delta {\cal H}}}({{\cal U} _S},{{\cal U} _T}) + 4\sqrt {\frac{{2{d _{vc}}\log (2N') + \log (2/\delta )}}{{N'}}}  + \lambda  .$$ Here, $ \lambda  = {{\epsilon} _T}({h^ * }) + {{\epsilon} _S}({h^ * }),{h^ * } = \arg {\min _{h \in {\cal H}}}{{\epsilon} _T}(h) + {{\epsilon} _S}(h)$.
>
> When $\lambda$ is small, the above bound achieves a convergence rate of $\sqrt{d _{vc} / N'} $. For large $\lambda$, it indicates that well-generalized classifiers cannot be learned under such data conditions. Our constant plays a similar role, representing the assumed ideal stability of the models. Here, requiring models to perform well on both datasets $D$ and $D^v$ is unnecessary, as this involves a tradeoff: the minimizers on both datasets may not achieve the best stability, and the best stability does not necessarily provide minimal risks on both datasets. This observation is essentially aligned with the nature of DG, since models that generalize well across both domains may not be minimizers on any single domain. Therefore, the existence of $h^*$ in our bounds is necessary.
>
> [1] Ben-David S, Blitzer J, Crammer K, Kulesza A, Pereira F, Vaughan JW. A theory of learning from different domains. Machine learning. 2010 May;79(1):151-75.

---

### Official Review · Reviewer_DCiy · 2025-11-02

**Soundness:** 3
**Presentation:** 3
**Contribution:** 3
**Rating:** 4
**Confidence:** 3

**Summary:**

This paper addresses the less-sample problem in domain generalization (DG), where limited training samples per domain lead to unstable model estimation and degraded generalization. To overcome the limitations of classical VC-bound–based generalization analysis (which assumes i.i.d. data), the authors propose a new stability-based theoretical framework using McDiarmid’s inequality to derive an exponential-decay generalization bound under non-i.i.d. data.
Building on this theory, the authors propose a regularization-based DG method, named DLAERM (Domain-Level Adaptive Empirical Risk Minimization), that adaptively controls the regularization strength λ according to data size and domain count. They also prove the method’s β-stability bound and Q-linear convergence, showing that stronger regularization improves stability for small-sample regimes.
Empirically, the method is validated on PACS and OfficeHome datasets using ResNet-50 as the feature extractor. DLAERM and its variants (e.g., DLAFISH, DLAEQRM) consistently outperform existing IRM-, ERM-, and regularization-based baselines, particularly under smaller training datasets (OfficeHome).

**Strengths:**

The paper provides a rigorous derivation of generalization bounds based on stability theory rather than the classical VC-bound, making it suitable for non-i.i.d. DG settings. The use of McDiarmid’s inequality to relax the independence assumption is theoretically elegant and relevant.

The proposed DLAERM framework naturally generalizes multiple regularization-based DG methods (LASERM, DLAFISH, etc.) through the same stability-based perspective. This offers a unified explanation for how λ controls stability and generalization.

Experiments on PACS (large sample) and OfficeHome (small sample) clearly show that DLA-based methods improve classification accuracy, demonstrating robustness to data scarcity.

**Weaknesses:**

Only two datasets (PACS, OfficeHome) are used, both for image classification. The method’s applicability to more complex DG benchmarks (e.g., TerraIncognita, VLCS, DomainNet) is not explored.

Although λ plays a central theoretical role, there is no systematic study of how varying λ affects stability, generalization gap, or convergence speed in practice.

Since the paper claims to solve small-sample problems, it would be insightful to compare with sample-efficient or augmentation-based methods (e.g., Mixup) that target similar challenges.

**Questions:**

How sensitive is the method to the choice of λ? Could an adaptive or learnable λ further improve performance?

How does the DLAERM framework behave when the number of domains E is very small (e.g., 2 or 3)? The bound suggests dependence on E, but experiments seem to focus on 4-domain setups.

Is there any trade-off observed between stability (small β) and model expressiveness (underfitting risk) when λ is too large?

---

> ### Author Response · Authors · 2025-11-20
>
> Thank you for taking the time to review our work. Your comments are extremely valuable for improving it, and we greatly appreciate your careful scrutiny.
>
> We greatly appreciate your suggestions regarding conducting experiments on more complex DG benchmarks, studying variations in $\lambda$, and comparing with augmentation-based methods. From the reviewers’ comments, we realize that our experimental evaluation is indeed limited. Our initial focus was primarily on theoretical derivations, which led us to underestimate the importance of comprehensive empirical validation and rely only on simple experiments. Thus, we conduct experiments on (1) synthetic data to validate the stability bounds $\beta$ and the learning convergence rate by varying $E$, $N$, and $\lambda$, and (2) real-world data for additional validation, including experiments with transformer-based backbones and with large datasets where $E$ and $N$ are manually controlled. We will update the results during the rebuttal phase to the best of our ability.
>
> We now address your questions as follows:
> 1. The choice of $\lambda$ in our bounds is based on the number of data samples. However, when applying the method to real-world data, another factor must be considered: the selected model. For example, in a CNN, a large $\lambda$ can induce strong inter-domain sparsity in the network weights, as discussed in our paper. If this sparsity is too high, the model may fail to approximate any function effectively. Therefore, the sensitivity of the choice of $\lambda$ is linked to $E$, $N$, and the selected model. Regarding adaptive or learnable methods for selecting $\lambda$, we cannot provide a definitive answer due to our limited knowledge in this area. We need to further exploration.
> 2. This is a very insightful question. We are aware that this is one limitation of our experiments, and we thank you for bringing it up. At this moment, we cannot fully address this question.
> 3. Yes, there is a tradeoff, as we mentioned in response to the first question. A too-large $\lambda$ may lead to the ineffectiveness of the trained models.
>
> Finally, we sincerely thank you again for your valuable suggestions regarding the experiments. These suggestions will help strengthen the claims in our work.

---

> ### Author Response · Authors · 2025-11-25
>
> Dear Reviewer,
>
> We have updated the experiments in the main paper, presenting results on synthetic data that are frequently used in domain generalization studies, for validating the stability bounds $\beta$ and analyzing learning convergence. Due to our limited experimental resources, the additional validations on real-world datasets cannot be provided during this rebuttal phase. Despite this, the flexibility of synthetic data in controlling $E$ and $N$ enables us to provide meaningful observations.
>
> The experimental results clearly exhibit: (1) the inverse proportional relationship between $\lambda$ and $EN$, (2) the optimal stability leading to good convergence, (3) the effectiveness of our method in addressing the less-sample problem in domain generalization, and (4) the tradeoff point for $\lambda$. All these results reinforce the significance of our contributions to domain generalization and address the limitations of our initial experiments.
>
> Now, we can fully answer your questions. (1) The learnable or adaptive $\lambda$ can improve performance if the corresponding result approaches the minimal point. Since the function for computing $\beta$ is convex, there is a minimal point. Such minimal stability leads to good learning convergence, which results in well-performing models. (2) DLAERM remains robust even when $E$ and $N$ are extremely small, such as $E=3$ and $N=10$. (3) The tradeoff point for $\lambda$ has been demonstrated in our new results, serving as the optimal stability.
>
> We hope you can review this revised version and provide us with further suggestions.
>
>
> Best,
>
> Authors.

---

### Official Review · Reviewer_Qtx9 · 2025-11-03

**Soundness:** 3
**Presentation:** 3
**Contribution:** 3
**Rating:** 6
**Confidence:** 3

**Summary:**

The paper develops a new theoretical framework to analyze and mitigate instability in \textbf{domain generalization (DG)} when training data per domain are scarce.
The authors argue that traditional VC-based generalization bounds are ineffective in DG since data are not i.i.d., and they instead derive stability-based exponential-decay generalization bounds using McDiarmid’s inequality.
Their main results show that for regularization-based learning methods, the generalization gap satisfies $\varepsilon \leq \mathcal{O}\left(\frac{1}{\sqrt{\lambda E N}}\right)$,
where $\lambda$ is the regularization factor, $E$ the number of domains, and $N$ the number of samples per domain.  They extend these results to learning in a reproducing kernel Hilbert space (RKHS) and demonstrate how the bounds motivate a new learning method, **DLAERM** (Domain-Level Aggregated Empirical Risk Minimization), which uses inter-domain sparsity to achieve stable generalization without requiring many domains.

Empirical evaluations on PACS and OfficeHome datasets with ResNet-50 validate the theoretical insight: DLAERM and its variants consistently outperform classical ERM and IRM-based baselines, especially in low-data regimes.

**Strengths:**

**Originality:**
1.  Introduces a stability-based theoretical framework tailored to domain generalization, extending Bousquet and Elisseeff’s stability notion beyond the i.i.d. setting.
2.  Derives exponential-decay generalization bounds using McDiarmid’s inequality, linking data size, regularization strength, and generalization gap in a novel way.
3. Proposes the DLAERM algorithm, which integrates theoretical insights into a practical learning objective with inter-domain sparsity.

**Quality:**
1.  The theoretical analysis is rigorous and carefully derived, with clear proofs and explicit assumptions (e.g., $\sigma$-admissibility, RKHS setup, convexity).
2.  Empirical results, though limited in scope, support the theoretical claims—especially the inverse proportionality between $\lambda$ and $EN$.
3.  The paper clearly articulates differences from prior work (VC-bounds, Ben-David et al. 2010, Bousquet \& Elisseeff 2002) and justifies the relevance of its assumptions.


**Clarity:**
1. The structure of the paper is logical and systematic: from motivation to theory, learning method, and experiments.
2.  Mathematical exposition is detailed yet readable, with intuitive remarks following each theorem and corollary.
3.  Figures and tables (especially Table 1 and Figure 1) effectively summarize empirical trends.


**Significance:**
    1. Provides a theoretical foundation for DG under small data, a highly relevant but under-explored regime.
    2. Offers an interpretable link between stability theory and practical DG learning algorithms.
    3. May inspire future research on stability-based regularization or generalization analysis in non-i.i.d. settings.

**Weaknesses:**

**Limited empirical validation.** Experiments are confined to two datasets and one backbone (ResNet-50). Further tests on non-vision tasks (e.g., text DG) or larger architectures would strengthen claims of generality.

**Comparative scope.** The experiments mainly modify existing baselines by adding the DLA regularizer. Stronger DG benchmarks or recent transformer-based backbones (e.g., ViT) are missing.

**Questions:**

1. Theorem 2 and Corollary 1 depend critically on the assumption that data across domains are independent. How sensitive are the stability bounds to mild violations of this assumption (e.g., correlated domain sampling)?


2.  In practice, how should $\lambda$ be tuned when both $E$ and $N$ vary? Is there an adaptive rule (e.g., $\lambda \propto 1/(EN)$) that matches empirical results?

3. Given that $\beta*$ can dominate the bound when large, is there a measurable or estimable way to assess $\beta*$ in real datasets?

---

> ### Author Response · Authors · 2025-11-20
>
> Thank you for taking the time to review our work. Your comments are extremely valuable for improving it, and we greatly appreciate your careful scrutiny.
>
> We greatly appreciate your suggestions regarding conducting experiments on non-vision tasks or other datasets, as well as incorporating additional network backbones. From the reviewers’ comments, we realize that our experimental evaluation is indeed limited. Our initial focus was primarily on theoretical derivations, which led us to underestimate the importance of comprehensive empirical validation and rely only on simple experiments. Thus, we conduct experiments on (1) synthetic data to validate the stability bounds $\beta$ and the learning convergence rate by varying $E$, $N$, and $\lambda$, and (2) real-world data for additional validation, including experiments with transformer-based backbones and with large datasets where $E$ and $N$ are manually controlled. We will update the results during the rebuttal phase to the best of our ability.
>
> We now address your questions as follows:
>
> 1. Stability bounds alone remain valid even when the data are not i.i.d., but the generalization bounds derived from stability do not. This is an insightful question, and after carefully examining the relevant content and proofs, we found that the stability bounds are indeed unaffected by the data distribution or dependence, which follows directly from Definition 1 and Lemma 1. In contrast, the independence of the data plays a crucial role in generalization bounds, since Theorem 1 relies on a McDiarmid-type inequality. Once the data are no longer independent, the bound in Theorem 1 fails to hold, and consequently the subsequent generalization bounds also fail.
> 2. If both $E$ and $N$ vary, we may treat $EN$ as a single quantity, i.e., $E \times N$, since in practice we typically focus on the total number of data samples in the given datasets. This naturally raises the question of how changes in $\lambda$, $E$, and $N$ jointly affect the results. Your question highlights the need to examine these factors experimentally, so thank you for bringing it up. Regarding the question on an adaptive rule, we cannot provide a complete response due to our limited knowledge of hyperparameter adaptation methods. According to Vapnik’s work [1], the less-sample problem refers to $EN/d _{vc} < 20$, and we may consider using a threshold-based approach to set $\lambda$ according to this criterion. However, $d _{vc}$ for deep models is difficult to define precisely. Therefore, we need to further explore experimental methods to adequately address your concern.
> 3. To estimate $\beta*$, we cannot fully guarantee feasibility. This best stability $\beta*$ requires the existence of $h* = \arg {\min _{h' \in {\cal H}}}\left| {R(h') - R'(h')} \right|$. This minimizer $ h* $ is assumed in our work. If we consider conducting experiments on synthetic data as in [2], it may be possible to estimate $\beta*$. However, for real-world datasets in domain generalization, the first step of obtaining $ h* $ is not available, and we cannot replace the ground-truth labels $Y$ in the datasets with $ h* $, since the conditions of labeling noise are also unknown. The existence of labeling noise implies that $ {\hat h} (X) \ne Y$, where $ {\hat h}$ is the minimizer for $R^{exp}(h,D)$. Nevertheless, if we can obtain such $h* $ on the given datasets using large models, then estimating $\beta*$ becomes feasible. Your question suggests an interesting direction for future work on leveraging large models, so thank you for bringing it up.
>
> Finally, we sincerely thank you again for these valuable suggestions.
>
> [1] Vapnik V. The nature of statistical learning theory. Springer science & business media; 2013 Jun 29.
>
> [2] Arjovsky M, Bottou L, Gulrajani I, Lopez-Paz D. Invariant risk minimization. arXiv preprint arXiv:1907.02893. 2019 Jul 5.

---

> > ### Comment · Reviewer_Qtx9 · 2025-11-21
> > **Thanks for the response**
> >
> > Thanks for the response, I will keep my positive score.

---

> > > ### Author Response · Authors · 2025-11-25
> > >
> > > Dear Reviewer,
> > >
> > > We have updated the experiments in the main paper, presenting results on synthetic data that are frequently used in domain generalization studies, for validating the stability bounds $\beta$ and analyzing learning convergence. Due to our limited experimental resources, the additional validations on real-world datasets cannot be provided during this rebuttal phase. Despite this, the flexibility of synthetic data in controlling $E$ and $N$ enables us to provide meaningful observations.
> > >
> > > The experimental results clearly exhibit: (1) the inverse proportional relationship between $\lambda$ and $EN$, (2) the optimal stability leading to good convergence, (3) the effectiveness of our method in addressing the less-sample problem in domain generalization, and (4) the tradeoff point for $\lambda$. All these results reinforce the significance of our contributions to domain generalization and address the limitations of our initial experiments.
> > >
> > > Now, we can fully answer your second question. We have a bound for $\lambda$ as an adaptive rule, i.e., $$ \lambda  \le \frac{C}{{\Delta \beta {E^2}N}},$$ where $ \Delta \beta  = \beta  - {\beta ^ * }$. Based on this, we can approximately estimate how to set $\lambda$, since the constant $C$ and the target stability $\Delta \beta$ should be chosen according to the specific task and dataset. This rule has been validated through the new experiments. Moreover, since a too large $\lambda$ could lead to failure in predictions of the learned model, this bound on $\lambda$ indicates a tradeoff point, which is also demonstrated in these new experiments.
> > >
> > > We hope you can review this revised version and provide us with further suggestions.
> > >
> > >
> > >
> > >
> > > Best,
> > >
> > > Authors.

---

### Author Response · Authors · 2025-11-29

Summary:


We first would like to thank the PCs, SACs, ACs, and Reviewers for their valuable time in reviewing our manuscript, as well as for their suggestions provided to improve our work. In particular, we appreciate Reviewer mfmp for offering a careful examination of the proofs of our main theoretical results.

Based on the initial round of comments, the main limitation of our work concerns the validation experiments (see comments from Reviewers Qtx9, DCiy, and mfmp), whereas Reviewer oqPD considered the novelty to be limited by the tools used to derive our theoretical results. For the experiments, we performed the relevant validations, updated the manuscript, and addressed the reviewers’ questions. Regarding the novelty, we argue it is fundamentally tied to the approach used to address the main problem rather than to the specific tools employed. Thus, we maintain our position in response to the comments from Reviewer oqPD.

During this rebuttal phase, we discussed only with two reviewers (Reviewers Qtx9 and mfmp), and they provided positive scores after the revision. However, we did not receive any response from the other two reviewers (Reviewers DCiy and oqPD), which is unfortunate for us.

Finally, we want to thank ICLR for providing a platform with a strong academic atmosphere.

---

### Meta-Review · Area_Chair_iuEZ · 2026-01-07

**Summary:**

The reviewers raised concerns about limited novelty, strong assumptions, insufficient experimental evaluation, and a mismatch between theory and practice. During the rebuttal, some minor issues were addressed; however, the main concerns—specifically the strong assumptions and insufficient experiments—remain unresolved. Therefore, I recommend rejecting this paper.

**Reviewer Concerns:**

Addressed:
- Reviewer mfmp: Missing discussion

Outstanding:
- Reviewer Qtx9: Limited empirical validation. Comparative scope.
- Reviewer DCiy: Insufficient experiments and ablation studies.
- Reviewer oqPD: the limited novelty, the strong assumptions, and insufficient experiments.

**Reviewer Scores:**

- Reviewer mfmp: Yes
- Reviewer Qtx9: No
- Reviewer DCiy: No
- Reviewer oqPD: No

---

### Decision · Program_Chairs · 2026-01-26

Reject